# Endoplasmic reticulum tubules limit the size of misfolded protein condensates

**Smriti Parashar[1], Ravi Chidambaram[1], Shuliang Chen[1], Christina R Liem[2], Eric Griffis[3], Gerard G Lambert[4], Nathan C Shaner[4], Matthew Wortham[5], Jesse C Hay[6], Susan Ferro-Novick[1]\***

[1]Department of Cellular and Molecular Medicine, University of California at San Diego, La Jolla, California, United States; [2]Division of Biological Sciences, University of California at San Diego, La Jolla, California, United States; [3]Nikon Imaging Center, University of California at San Diego, La Jolla, California, United States; [4]Department of Neurosciences, University of California at San Diego, La Jolla, California, United States; [5]Department of Pediatrics, Pediatric Diabetes Research Center, University of California at San Diego, La Jolla, California, United States; [6]Division of Biological Sciences and Center for Structural & Functional Neuroscience, University of Montana, Missoula, United States

**\*For correspondence:**
sfnovick@ucsd.edu

**Competing interest:** The authors declare that no competing interests exist.

**Abstract** The endoplasmic reticulum (ER) is composed of sheets and tubules. Here we report that the COPII coat subunit, SEC24C, works with the long form of the tubular ER-phagy receptor, RTN3, to target dominant-interfering mutant proinsulin *Akita* puncta to lysosomes. When the delivery of *Akita* puncta to lysosomes was disrupted, large puncta accumulated in the ER. Unexpectedly, photobleach analysis indicated that *Akita* puncta behaved as condensates and not aggregates, as previously suggested. *Akita* puncta enlarged when either RTN3 or SEC24C were depleted, or when ER sheets were proliferated by either knocking out Lunapark or overexpressing CLIMP63. Other ER-phagy substrates that are segregated into tubules behaved like *Akita*, while a substrate (type I procollagen) that is degraded by the ER-phagy sheets receptor, FAM134B, did not. Conversely, when ER tubules were augmented in Lunapark knock-out cells by overexpressing reticulons, ER-phagy increased and the number of large *Akita* puncta was reduced. Our findings imply that segregating cargoes into tubules has two beneficial roles. First, it localizes mutant misfolded proteins, the receptor, and SEC24C to the same ER domain. Second, physically restraining condensates within tubules, before they undergo ER-phagy, prevents them from enlarging and impacting cell health.

## Introduction

The endoplasmic reticulum (ER) forms a continuous polygonal network of interconnected sheets and tubules that extends from the nucleus to the cell periphery (*Chen et al., 2013*). This unique, evolutionarily conserved, architecture has been extensively studied at a morphological and biochemical level, yet the functional advantage conferred by these distinct domains remains unclear (*Shibata et al., 2006*). ER shape is dependent on the reticulons (RTN), atlastin (ATL), and lunapark (LNPK) proteins (*Voeltz et al., 2006*; *De Craene et al., 2006*; *Hu et al., 2009*; *Orso et al., 2009*; *Chen et al., 2012*; *Chen et al., 2015*). The reticulons are required to stabilize tubules, while the atlastins mediate tubule-tubule fusion (*Voeltz et al., 2006*; *De Craene et al., 2006*; *Hu et al., 2009*; *Orso et al., 2009*). LNPK, which resides at the junctions formed when two tubules fuse, regulates ER architecture by stabilizing nascent tubule junctions (*Chen et al., 2015*). In the absence of LNPK, ER junctions are destabilized, and as a consequence, the tubular network collapses (*Chen et al., 2015*). ER network

organization may be important for cell health as mutations in ER-shaping proteins have been associated with neurodegenerative disorders, such as hereditary sensory autonomic neuropathy and hereditary spastic paraplegias (*Zhang and Hu, 2016*).

A major function of the ER is protein biogenesis. When errors in protein processing, protein folding, and protein complex assembly occur, ER proteostasis is disrupted and aberrant proteins accumulate in the ER (*Sun and Brodsky, 2019*). Cells use two major degradative mechanisms to restore homeostasis, the proteasome and autophagy (*Sun and Brodsky, 2019*). Misfolded ER proteins can be retrotranslocated across the membrane into the cytosol by the ER-associated degradation (ERAD) machinery where they are degraded by the proteasome (*Sun and Brodsky, 2019*). Some misfolded proteins, however, are resistant to ERAD and must use other disposal pathways (*Sun and Brodsky, 2019*). Autophagy is an alternate pathway that degrades organelles, pathogens, protein aggregates, and liquid protein condensates (*Mizushima et al., 2011*; *Yamasaki et al., 2020*). ER autophagy (termed ER-phagy) is a type of selective autophagy that targets a domain of the ER to the lysosome for degradation (*Fregno and Molinari, 2019*; *Wilkinson, 2020*; *Chino and Mizushima, 2020*; *Hübner and Dikic, 2020*; *Ferro-Novick et al., 2021*). ER to lysosome degradation occurs via a double membrane vesicle called the autophagosome (macro-ER-phagy) or can be non-autophagosome mediated (*Fregno and Molinari, 2019*; *Wilkinson, 2020*; *Chino and Mizushima, 2020*; *Hübner and Dikic, 2020*; *Ferro-Novick et al., 2021*). ER-phagy uses receptors that connect the ER to the autophagy machinery via their ability to bind Atg8 in yeast or LC3 or GABARAP in mammals (*Fregno and Molinari, 2019*; *Wilkinson, 2020*; *Chino and Mizushima, 2020*; *Hübner and Dikic, 2020*; *Ferro-Novick et al., 2021*). To date, two ER-phagy membrane receptors have been identified in yeast and six in mammals. Additionally, several soluble receptors have recently been reported (*Ferro-Novick et al., 2021*). Different membrane ER-phagy receptors mark the distinct morphological domains of the ER. For example, in mammals, the autophagy receptor FAM134B resides in the curved edges of ER sheets, while the long form of the ER-phagy receptor RTN3 (herein called RTN3) localizes to the tubular ER (*Khaminets et al., 2015*; *Grumati et al., 2017*). Recent studies have revealed that FAM134B and RTN3 are required for ER proteostasis (*Forrester et al., 2019*; *Schultz et al., 2018*; *Fregno et al., 2018*; *Cunningham et al., 2019*). These receptors contain reticulon homology domains that generate regions of high membrane curvature and have ER fragmenting activity (*Khaminets et al., 2015*; *Grumati et al., 2017*).

The delivery of ER tubules to lysosomes during ER-phagy also requires SEC24C in mammals and Lst1 in yeast (*Cui et al., 2019*). SEC24C, and its homologue Lst1, are COPII coat cargo adaptors that sort proteins into ER-derived transport carriers that traffic to the Golgi (*Zanetti et al., 2011*; *Gomez-Navarro and Miller, 2016*). Here we have asked if SEC24C is also required for the targeting of mutant misfolded cargoes to lysosomes during ER-phagy. For our studies, we analyzed mutant proinsulin *Akita*, as well as misfolded prohormone pro-opiomelanocortin (C28F POMC) and the mutant neuropeptide pro-arginine-vasopressin (G57S Pro-AVP). All three cargoes are dominant-interfering disease-causing proteins that are segregated into tubules before undergoing ER-phagy. In addition, we have used different methods to shift the distribution of ER between sheets and tubules and asked if restricting cargoes to tubules plays a role in ER proteostasis.

## Results

### SEC24C colocalizes with LC3B and RTN3 in Torin-treated cells

We previously showed that when autophagy is induced in *Saccharomyces cerevisiae* with the TORC1 kinase inhibitor, rapamycin, Lst1 associates with Atg8 and Atg40 to facilitate the packaging of ER into autophagosomes (*Cui et al., 2019*). We named the sites on the ER where Lst1 colocalizes with the autophagy machinery *ER-ph*agy *s*ites (ERPHS) (*Cui et al., 2019*). To begin to address if ERPHS form on the mammalian ER, we asked if SEC24C colocalizes with LC3B in U2OS cells treated with the TORC1 inhibitor, Torin 2. As shown in *Figure 1A and B*, a significant increase in the colocalization of SEC24C with LC3B was observed when ER-phagy was induced with Torin 2. SEC24C was also delivered to lysosomes during Torin treatment (*Figure 1—figure supplement 1A and B*), and transport was blocked with MRT68921, an inhibitor of the ULK1/2 kinase that disrupts autophagosome formation (*Petherick et al., 2015*).

Next, we wanted to identify the ER-phagy receptor that acts with SEC24C. FAM134B and RTN3 were the most obvious candidates as yeast Atg40 is related to both receptors (*Cui et al., 2019*;

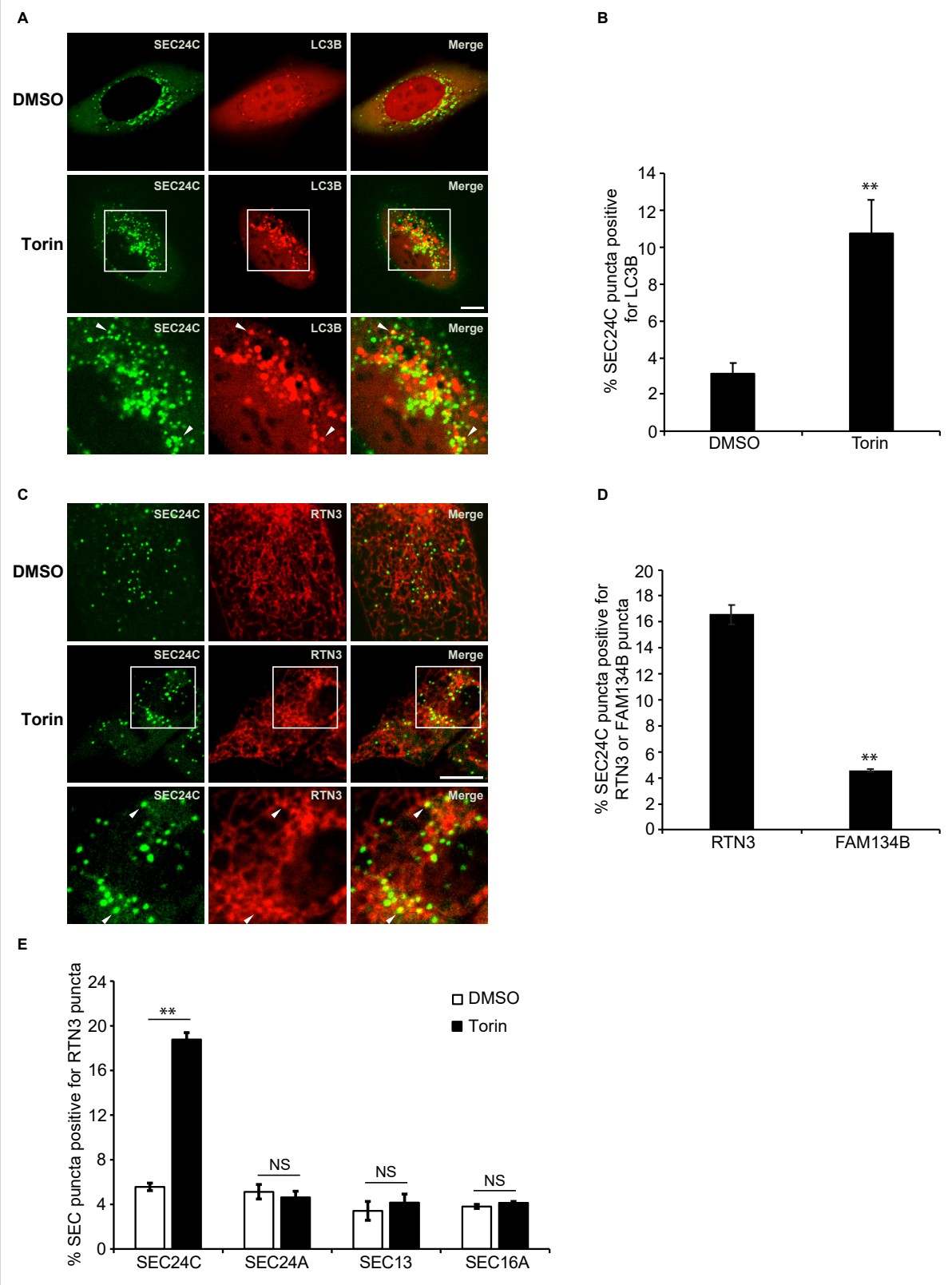

**Figure 1.** Torin induces the colocalization of SEC24C with LC3B and RTN3, but not FAM134B. (**A**) U2OS cells expressing EYFP-SEC24C and mCherry-LC3B were treated with Torin 2 for 3.5 hr and examined by confocal microscopy. Arrowheads in the inset indicate SEC24C puncta colocalizing with LC3B. (**B**) Bar graph showing the % of EYFP-SEC24C puncta colocalizing with mCherry-LC3B puncta for the data shown in (**A**). (**C**) Cells expressing mCherry-RTN3 and EYFP-SEC24C were treated with Torin 2 for 3.5 hr. Representative confocal images are shown. Arrowheads in the inset indicate EYFP-SEC24C

*Figure 1 continued on next page*

Figure 1 continued

puncta colocalizing with mCherry-RTN3. (**D**) Bar graph showing the % of EYFP-SEC24C puncta colocalizing with mCherry-RTN3 or FAM134B-mTurquoise puncta in transiently transfected cells. (**E**) Cells stably expressing mCherry-RTN3 were transfected with EYFP-SEC24C, mAaus0.5-SEC24A, SEC13-GFP, or EGFP-SEC16A and treated with Torin 2 for 3.5 hr. The % of SEC puncta that colocalized with mCherry-RTN3 puncta was quantified. Scale bars in (**A**) and (**C**), 10 μm. Error bars in (**B**), (**D**), and (**E**) represent SEM; n = 3 independent experiments. Approximately 20–30 cells/experiment were analyzed. NS: not significant (p≥0.05); **p<0.01, Student's unpaired *t*-test.

The online version of this article includes the following figure supplement(s) for figure 1:

**Figure supplement 1.** SEC24C is delivered to lysosomes in the presence of Torin 2.

*Mochida et al., 2015*). The Atg40 domain structure is similar to FAM134B, yet Atg40 and RTN3 both localize to ER tubules (*Grumati et al., 2017*; *Mochida et al., 2015*). FAM134B, RTN3, and COPII coat subunits all reside on curved membranes (*Okamoto et al., 2012*). We found that SEC24C colocalized with RTN3, but not FAM134B, in Torin 2-treated cells (*Figure 1C and D*, *Figure 1—figure supplement 1C*). This finding is consistent with published mass spectrometry data and immunoprecipitation studies showing that RTN3, but not FAM134B, co-precipitates with SEC24C (*Grumati et al., 2017*). Furthermore, the long form of RTN3, but not the short form, specifically co-precipitates with the endogenous copy of SEC24C during starvation induced ER-phagy, and only the long form acts in ER-phagy (*Grumati et al., 2017*).

In addition to SEC24C, mammalian cells have three other SEC24 paralogs (*Zanetti et al., 2011*). SEC24A and SEC24B are 50% identical to each other and closely related to the major yeast secretory cargo adaptor, Sec24 (*Zanetti et al., 2011*; *Gomez-Navarro and Miller, 2016*). SEC24C and SEC24D are 50% identical to each other, but only weakly homologous to SEC24A and SEC24B (*Zanetti et al., 2011*). In yeast, Lst1, but not its paralog Sec24, colocalizes with Atg40 in the presence of rapamycin (*Cui et al., 2019*). The coat outer shell, Sec13-Sec31, also does not colocalize with Atg40 (*Cui et al., 2019*). Similar to what was observed in yeast, SEC24C, but not SEC24A or SEC13, colocalized with RTN3 in a Torin 2-dependent manner (*Figure 1E*, *Figure 1—figure supplement 1D*). These localization studies are also consistent with mass spectrometry data showing that only SEC24C, and not SEC24A, interacts with RTN3 (*Grumati et al., 2017*). Torin 2 also did not induce the colocalization of RTN3 with SEC16 (*Figure 1E*, *Figure 1—figure supplement 1D*), a marker for the *ER* exit sites (ERES) where secretory cargo leaves the ER (*Barlowe and Helenius, 2016*; *Maeda et al., 2019*). These findings show that the RTN3-SEC24C colocalizing sites, induced by Torin 2, are distinct from ERES and appear to be equivalent to yeast ERPHS (*Cui et al., 2019*). Thus, SEC24C-RTN3-mediated ER-phagy is evolutionarily conserved from yeast to man.

### *Akita*, but not hCOL1A1, puncta accumulate in SEC24C depleted cells

To identify and analyze cargo that utilizes SEC24C-mediated ER-phagy for clearance from the ER, we began with mutant proinsulin *Akita*, a substrate known to be degraded by RTN3-dependent ER-phagy (*Cunningham et al., 2019*). *Akita* causes an autosomal-dominant form of diabetes, mutant *INS-* gene-induced diabetes of youth (MIDY) (*Cunningham et al., 2019*). High-molecular-weight mutant oligomeric forms of *Akita* that cannot be cleared by ERAD are disposed of by RTN3, and not FAM134B (*Cunningham et al., 2019*). In the absence of RTN3, *Akita* fails to be delivered to lysosomes and large *Akita* puncta, which are thought to be aggregates, accumulate intracellularly (*Cunningham et al., 2019*; *Chen et al., 2020*). We found that the delivery of *Akita*-sfGFP to lysosomes was also dependent on SEC24C, but not SEC24A (*Figure 2A and B*, *Figure 2—figure supplement 1A*). Furthermore, large *Akita* puncta (≥0.5 μm$^2$) accumulated in cells depleted of SEC24C, but not SEC24A, SEC24B or SEC24D (*Figure 2C and D*, *Figure 2—figure supplement 1B*). The puncta that accumulated in the siSEC24C cells were similar in size to the large *Akita* puncta that accumulated in the siRTN3-depleted cells (*Figure 2—figure supplement 1C*). Additionally, the simultaneous knockdown of SEC24C and RTN3 did not enhance the accumulation of large *Akita* puncta, implying that SEC24C and RTN3 act on the same pathway (*Figure 2—figure supplement 1C and D*).

To compare these findings with a known FAM134B-mediated ER-phagy substrate, we examined type I procollagen (hCOL1A1). Previous studies have shown that FAM134B targets misfolded procollagens to lysosomes during ER-phagy (*Forrester et al., 2019*). Approximately 20% of newly synthesized type I procollagen is misfolded (*Bienkowski et al., 1986*). Although control cells contained some large puncta (≥0.5 μm$^2$) (*Figure 2—figure supplement 1E*), numerous small hCOL1A1 puncta

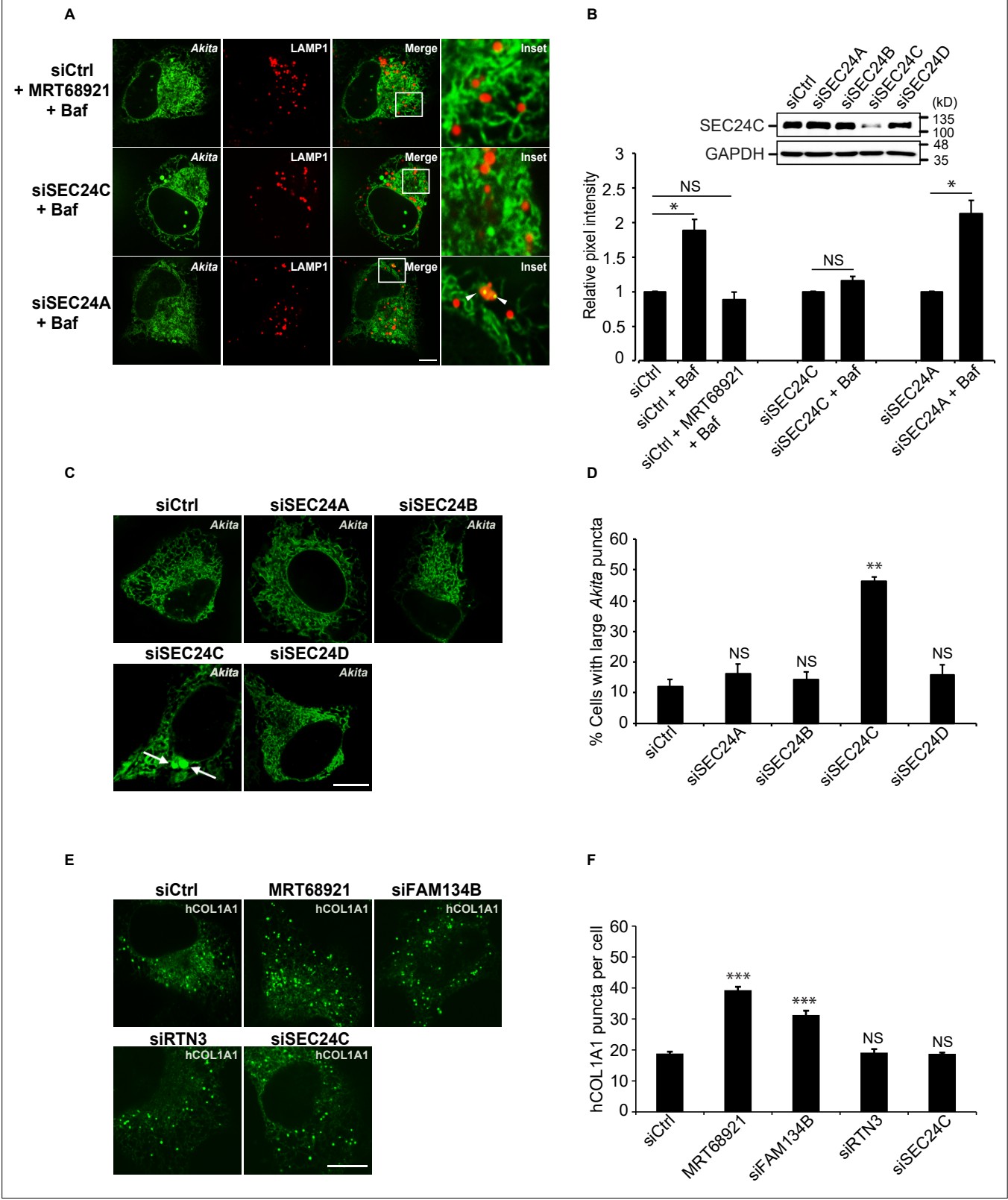

**Figure 2.** *Akita,* but not hCOL1A1, accumulates in the endoplasmic reticulum (ER) as large puncta in SEC24C-depleted cells. **A)** U2OS cells expressing *Akita*-sfGFP and LAMP1-mCherry were depleted of SEC24C or SEC24A by siRNA, and treated with bafilomycin A1 (Baf) before imaging. Arrowheads in the inset indicate *Akita* in the LAMP1 structures. (**B**) Quantitation of *Akita*-sfGFP in LAMP1-mCherry structures for the data shown in (**A**). The DMSO control for each condition was set to 1.0. The relative pixel intensity for each condition is the mean intensity of *Akita*-sfGFP in the pixels that overlap

*Figure 2 continued on next page*

*Figure 2 continued*

with LAMP1. In the top-right corner, the specificity of the SEC24C knockdown is shown. Cells were depleted of the different SEC24 isoforms by siRNA and immunoblotted for SEC24C. (**C**) Cells were depleted of the different SEC24 isoforms and analyzed for the accumulation of large *Akita* puncta ($\geq 0.5$ $\mu m^2$). Arrows point to large *Akita* puncta. (**D**) Bar graph showing the % of cells with large *Akita* puncta for the data shown in (**C**). Large *Akita* puncta only accumulated in siSEC24C cells; however, the % siSEC24C cells with puncta of all sizes ($51.7 \pm 3.3\%$) appeared to be roughly the same as the siCtrl ($55.3 \pm 1.1\%$). (**E**) Cells expressing EGFP-hCOL1A1 were treated for 3.5 hr with MRT68921 or depleted of FAM134B, RTN3, or SEC24C by siRNA and analyzed for the accumulation of EGFP-hCOL1A1 puncta. (**F**) Bar graph showing EGFP-hCOL1A1 puncta per cell for the data shown in (**E**). Puncta of all sizes were quantitated. Scale bars in (**A**), (**C**), and (**E**), 10 µm. Error bars in (**B**), (**D**), and (**F**) represent SEM; n = 3–4 independent experiments, n = 20–40 cells/experiment. NS: not significant ($p \geq 0.05$); *$p < 0.05$, **$p < 0.01$; ***$p < 0.001$, Student's unpaired *t*-test.

The online version of this article includes the following source data and figure supplement(s) for figure 2:

**Source data 1.** Uncropped blots for *Figure 2B*.

**Figure supplement 1.** The delivery of *Akita* to lysosomes requires SEC24C.

**Figure supplement 1—source data 1.** Uncropped blots for *Figure 2—figure supplement 1B*.

**Figure supplement 1—source data 2.** Uncropped blots for *Figure 2—figure supplement 1D*.

**Figure supplement 1—source data 3.** Uncropped blots for *Figure 2—figure supplement 1F*.

(generally <0.12 $\mu m^2$) were found in the majority of cells (*Figure 2E*). The smaller puncta (<0.5 $\mu m^2$) accumulated in the FAM134B-depleted and MRT68921-treated cells (*Figure 2E and F*), while cells with large puncta ($\geq 0.5$ $\mu m^2$) did not increase in number when FAM134B-mediated ER-phagy was disrupted (*Figure 2—figure supplement 1F*). Consistent with the proposal that SEC24C works with RTN3, and not FAM134B, hCOL1A1 puncta did not accumulate in the RTN3- or SEC24C-depleted cells (*Figure 2E–F*).

## SEC24C colocalizes with small, highly mobile *Akita* puncta

To ask if SEC24C associates with large *Akita* puncta ($\geq 0.5$ $\mu m^2$), colocalization studies were performed with cells treated from 0 to 6 hr with MRT68921 (*Figure 3A and B*). In addition to large puncta, small *Akita* puncta were also observed in treated as well as untreated cells (see size range in *Figure 3C and D*). Unexpectedly, we found that the small (<0.12 $\mu m^2$), but not the larger (>0.12 $\mu m^2$), puncta colocalized with SEC24C (*Figure 3C and D*). The small puncta were also associated with tubules and were frequently seen at tubule junctions, as well as the tips of tubules (*Video 1*). These puncta moved with the tubules as they branched and fused, and some puncta appeared to change in size and shape as they traveled through the network (*Video 1*). In contrast, the larger puncta were mostly seen in sheet-like dense regions of the ER and moved more slowly (*Video 2*, *Figure 3E*). While puncta movement inversely correlated with size (*Figure 3E*), no difference in puncta movement was observed in the absence or presence of MRT68921 (*Figure 3—figure supplement 1A*). Fluorescence recovery after photobleaching (FRAP) of the large *Akita* puncta revealed a fast recovery time ($T_{1/2}$ = ~13 s, *Figure 3F*; *Figure 3—figure supplement 1B–D*). The recovery of the large puncta was indistinguishable from the recovery of diffuse *Akita*-sfGFP in the ER network, indicating that the *Akita*-sfGFP in the puncta rapidly exchanged with the diffuse pool in the ER. In contrast, a substrate of FAM134B-mediated ER-phagy, EGFP-hCOL1A1, behaved differently. The fluorescence recovery of EGFP-hCOL1A1 puncta in siFAM134B cells was decreased (~60%) relative to diffuse EGFP-hCOL1A1 in the network (~90%) (*Figure 3—figure supplement 2A–D*). These findings are consistent with previous studies demonstrating that procollagen type 1 aggregates in the ER and show decreased fluorescence recovery after photobleaching (*Ishida et al., 2009*). In total, the ability of *Akita* to concentrate into mobile structures that rapidly exchanged their contents with the ER implies that *Akita* puncta are not aggregates, as previously suggested (*Cunningham et al., 2019*). Instead, these puncta behaved as liquid condensates (see criteria established in *Banani et al., 2017*). When ER-phagy was disrupted, the *Akita* puncta accumulated and enlarged in what appeared to be sheet-like regions of the ER.

## *Akita* colocalizes with SEC24C-SEC23A and LC3B

As no one has shown that misfolded cargo localizes to the ERPHS, we wanted to ask if the *Akita*-SEC24C colocalizing puncta are sites of ER-phagy. We found that the small *Akita* puncta preferentially colocalized with SEC24C, and not SEC24A (*Figure 4—figure supplement 1A and B*). The binding partner for all SEC24 isoforms, SEC23A, also colocalized with *Akita* (*Figure 4—figure supplement*

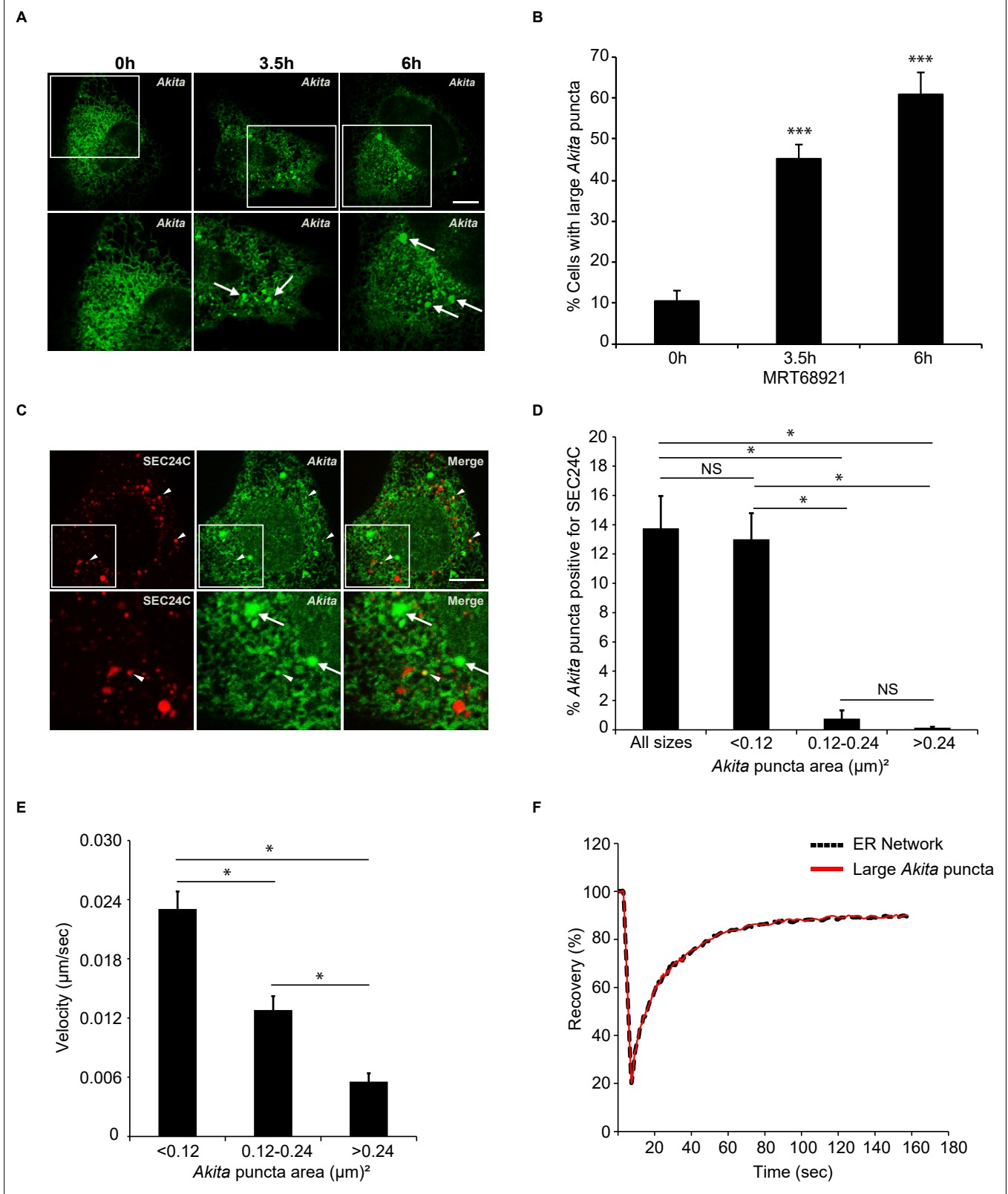

**Figure 3.** Highly mobile small *Akita* puncta colocalize with SEC24C. **A**) U2OS cells expressing *Akita*-sfGFP were treated with MRT68921 for the indicated times and examined for the accumulation of large puncta (≥0.5 μm²). Arrows mark large *Akita* puncta. (**B**) Bar graph showing the % of cells with large *Akita* puncta for the data shown in (**A**). (**C**) Cells expressing *Akita*-sfGFP and mCherry-SEC24C were treated with MRT68921 for 3.5 hr and imaged. Arrowheads show *Akita* puncta colocalizing with SEC24C, arrows indicate *Akita* puncta that do not colocalize with SEC24C. (**D**) Bar graph for the data

*Figure 3 continued on next page*

*Figure 3 continued*

shown in (**C**) of % *Akita*-sfGFP puncta of different sizes that colocalize with mCherry-SEC24C puncta. (**E**) Cells expressing *Akita*-sfGFP were treated with MRT68921 for 3.5 hr, and the velocity of differently sized puncta was determined. (**F**) Cells expressing *Akita*-sfGFP were treated with MRT68921 for 3.5 hr followed by fluorescence recovery after photobleaching (FRAP) analysis of *Akita* puncta (average size; ≥ 0.32 μm²) and the endoplasmic reticulum network. Scale bars in (**A**) and (**C**), 10 μm. Error bars in (**B**), (**D**), and (**E**) represent SEM; n = 3 independent experiments. Approximately 30–40 cells were examined/experiment (**B**), 20–30 cells/experiment (**D**), 35–40 puncta/experiment (**E**), and 20 puncta/experiment (**F**). NS: not significant (p≥0.05); *p<0.05, ***p<0.001, Student's unpaired *t*-test.

The online version of this article includes the following figure supplement(s) for figure 3:

**Figure supplement 1.** The small *Akita* puncta are highly mobile.

**Figure supplement 2.** hCOL1A1 puncta do not rapidly recover after photobleaching.

*1C and D*). To compare *Akita* with a cargo that leaves the ER from ERES, we examined proinsulin, the wild-type form of *Akita*. Consistent with published localization studies, we observed a juxtanuclear pool of proinsulin (*Figure 4—figure supplement 2A*; *Haataja et al., 2013*). The juxtanuclear steady-state pool was previously shown to colocalize with the early Golgi marker p115 in U2OS cells (*Haataja et al., 2013*). Additionally, we also observed proinsulin on ER tubules and on puncta that colocalize with the ERES marker, SEC24A (*Figure 4—figure supplement 2A and B*). In contrast to what we observed for *Akita*, proinsulin puncta colocalized equally as well with SEC24A and SEC24C (*Figure 4—figure supplement 2A and B* ). Interestingly, the proinsulin puncta that colocalized with SEC24A appeared to be similar in size to the *Akita* puncta that colocalized with SEC24C (*Figure 4—figure supplement 2C*).

If the *Akita* puncta that colocalized with SEC24C represent sites on the ER where ER-phagy is initiated, these puncta should increase in number when autophagosome formation is blocked. To ask if this occurs, we treated cells for 3.5 hr with MRT68921 and quantitated the *Akita* puncta that colocalize with SEC24C in the presence and absence of inhibitor. A dramatic increase in the percent of cells with multiple *Akita* puncta that colocalized with SEC24C was observed in the presence of inhibitor, while no significant increase was found for SEC24A (*Figure 4A and B*). These localization studies are in accord with the finding that SEC24A is not needed for the delivery of *Akita* to lysosomes (*Figure 2A and B*), and the observation that large *Akita* puncta do not accumulate in siSEC24A cells (*Figure 2C and D*).

The increased colocalization of *Akita* with SEC24C, observed during treatment with MRT68921, was dependent on RTN3 (*Figure 4C*, *Figure 4—figure supplement 3A*). When we analyzed the localization of LC3B, a known binding partner for RTN3 (*Grumati et al., 2017*), the number of cells with multiple *Akita* puncta colocalizing with LC3B also increased in the presence of MRT68921 (*Figure 4D and E*). Additionally, this increase was dependent on RTN3 (*Figure 4E*, *Figure 4—figure supplement 3B*). *Akita* puncta still colocalized with LC3B in the absence of SEC24C (*Figure 4—figure supplement 3C and D*), indicating that the association of RTN3 with LC3B does not depend on SEC24C. In total, these studies imply that small *Akita* puncta that associate with RTN3, LC3B, and SEC24C form ERPHS on ER tubules.

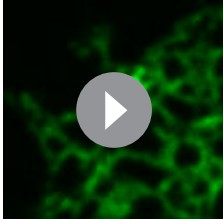

**Video 1.** Small *Akita* puncta rapidly move in the tubular endoplasmic reticulum (ER) network. Time-lapse images of a small *Akita*-sfGFP punctum (0.07 μm²) as it moves in the tubular network of U2OS cells. The punctum appears to change size and shape as it moves.

https://elifesciences.org/articles/71642/figures#video1

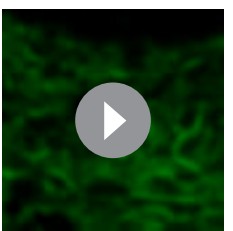

**Video 2.** Large *Akita* puncta accumulate in the dense endoplasmic reticulum (ER) and exhibit low mobility. Time-lapse images of a large *Akita*-sfGFP punctum (1.249 μm²) in a dense ER region of a cell that was treated with MRT68921 for 3.5 hr.

https://elifesciences.org/articles/71642/figures#video2

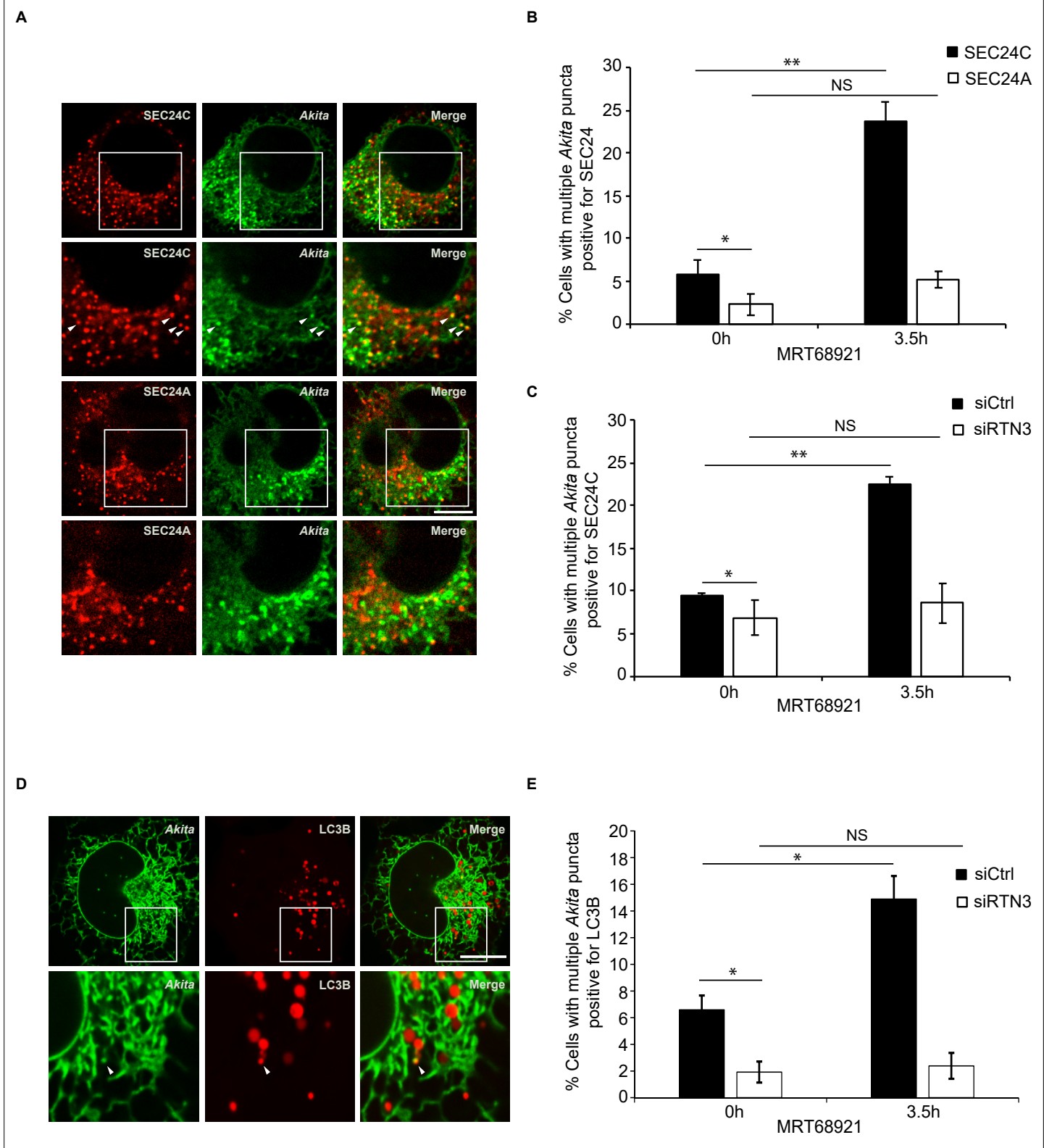

**Figure 4.** Disrupting autophagy leads to the accumulation of *Akita* puncta that colocalize with SEC24C, and LC3B, but not SEC24A. (**A**) U2OS cells expressing *Akita*-sfGFP and mCherry-SEC24C or mCherry-SEC24A were treated with MRT68921 for 3.5 hr. Arrowheads in the inset indicate *Akita* puncta colocalizing with SEC24C. (**B**) Bar graph showing the % of cells with multiple *Akita*-sfGFP puncta colocalizing with mCherry-SEC24C or mCherry-SEC24A puncta 0 or 3.5 hr after treatment with MRT68921. (**C**) Cells expressing *Akita*-sfGFP and mCherry-SEC24C were depleted of RTN3 by RNAi and

*Figure 4 continued on next page*

Figure 4 continued

treated with MRT68921 for 0 or 3.5 hr. The % of cells showing multiple *Akita*-sfGFP puncta colocalizing with mCherry-SEC24C puncta was quantified at the indicated time points. (**D**) Cells expressing *Akita*-sfGFP and mCherry-LC3B were depleted of RTN3 and treated with MRT68921 for 0 or 3.5 hr. A representative image for control cells (3.5 hr) is shown. Arrowhead in the inset shows an *Akita* puncta colocalizing with LC3B. (**E**) Bar graph showing the % of cells with multiple *Akita*-sfGFP puncta colocalizing with mCherry-LC3B puncta for the data shown in (**D**). Scale bars in (**A**) and (**D**), 10 µm. Error bars in (**B**), (**C**), and (**E**) represent SEM; n = 3 independent experiments. Approximately 20–30 cells/experiment were examined. NS: not significant (p≥0.05); *p<0.05, **p<0.01, Student's unpaired *t*-test.

The online version of this article includes the following figure supplement(s) for figure 4:

**Figure supplement 1.** *Akita* also colocalizes with SEC23A.

**Figure supplement 2.** Proinsulin colocalizes with both SEC24C and SEC24A.

**Figure supplement 3.** *Akita* fails to colocalize with SEC24C, and LC3B, in the absence of RTN3.

## *Akita* puncta enlarge in sheet-like ER

The studies discussed above suggest that small *Akita* puncta reside in tubules, while the larger puncta appear to be in sheet-like ER. To determine the effect on *Akita* of shifting the distribution of ER from tubules to sheets, we analyzed puncta in LNPK KO cells (*Wang et al., 2016*). In the absence of LNPK, fewer junctions are observed and the tubular network collapses (*Chen et al., 2012*; *Chen et al., 2015*). The loss of tubules in the knock-out (KO) cells shifts the morphology of the ER to a more sheet-like network (*Chen et al., 2015*). We found that ~76% of the LNPK KO cells contained large puncta (*Figure 5A*), while ~15% of the control cells and ~45% of the siRTN3 and siSEC24C-depleted cells contained large puncta (*Figure 2D*, *Figure 2—figure supplement 1C*). A large fraction (~35%) of the LNPK KO cells (*Figure 5A*) also contained multiple large puncta. To examine the *Akita* puncta in more detail, we analyzed 3D reconstructions of deconvolved Z stacks of control and LNPK KO cells (see representative *Videos 3 and 4*, and frames in *Figure 5B and C*). This analysis verified that the small puncta reside in tubules, while the large puncta are in ER sheets. To show by a second method that *Akita* puncta enlarge in ER sheets, we proliferated sheets by overexpressing the sheet producing protein, CLIMP63 (*Shibata et al., 2010*). ER sheet proliferation, which was induced by CLIMP63 over-expression, increased the percentage of cells with large *Akita* puncta (*Figure 5—figure supplement 1A–D*, *Video 5*), as well as the mean size of *Akita* puncta per cell (*Figure 5—figure supplement 1E*). A similar increase in the mean size of *Akita* puncta per cell was also observed in LNPK KO cells (*Figure 5—figure supplement 1F*). The co-overexpression of CLIMP63 with RTN4, an ER protein that drives ER tubulation (*Wang et al., 2016*), decreased the percentage of cells with sheets as well as the number of large puncta (*Figure 5—figure supplement 1C–E*, *Video 6*). RTN4 overexpression also decreased the mean size of *Akita* puncta per cell in LNPK KO cells (*Figure 5—figure supplement 1F*). Consistent with the accumulation of large *Akita* puncta in the ER, a decrease in the delivery of *Akita* to lysosomes was observed in LNPK KO and CLIMP63 overexpressing cells (*Figure 5—figure supplement 2A and B*). Together these findings indicate that *Akita* puncta enlarge when ER sheets proliferate.

Next, we examined LNPK KO cells expressing hCOL1A1, a cargo that is targeted to ER-phagy by the ER sheets receptor, FAM134B (*Forrester et al., 2019*). In contrast to what was observed for *Akita*, we did not see an increased accumulation of hCOL1A1 puncta in LNPK KO cells (*Figure 5—figure supplement 3A and B*). This was true for the small (*Figure 5—figure supplement 3A*) as well as the large hCOL1A1 puncta (*Figure 5—figure supplement 3B*). Because hCOL1A1 behaved differently than *Akita* in LNPK KO cells, we also analyzed two other cargoes whose degradation is known to be dependent on RTN3; G57S mutation of pro-arginine-vasopressin (G57S Pro-AVP), and the C28F mutation in pro-opiomelanocortin (C28F POMC). G57S Pro-AVP is a mutant neuropeptide whose expression leads to an autosomal-dominant neurodegenerative disorder called neurohypophyseal diabetes insipidus (DI) (*Spiess et al., 2020*). C28F POMC is a mutant prohormone that causes early onset obesity (*Kim et al., 2018*; *Creemers et al., 2008*). Like *Akita*, large puncta of G57S Pro-AVP accumulated in the siSEC24C, but not siSEC24A, depleted cells (*Figure 5—figure supplement 3C*). Furthermore, large puncta accumulation was more pronounced in LNPKO cells (*Figure 5—figure supplement 3D*). Similar results were obtained with C28F POMC (*Figure 5—figure supplement 3E and F*). In addition, a significant fraction (>30%) of the LNPK KO cells that contained G57S Pro-AVP (*Figure 5—figure supplement 3D*) or C28F POMC (*Figure 5—figure supplement 3F*) contained multiple large puncta.

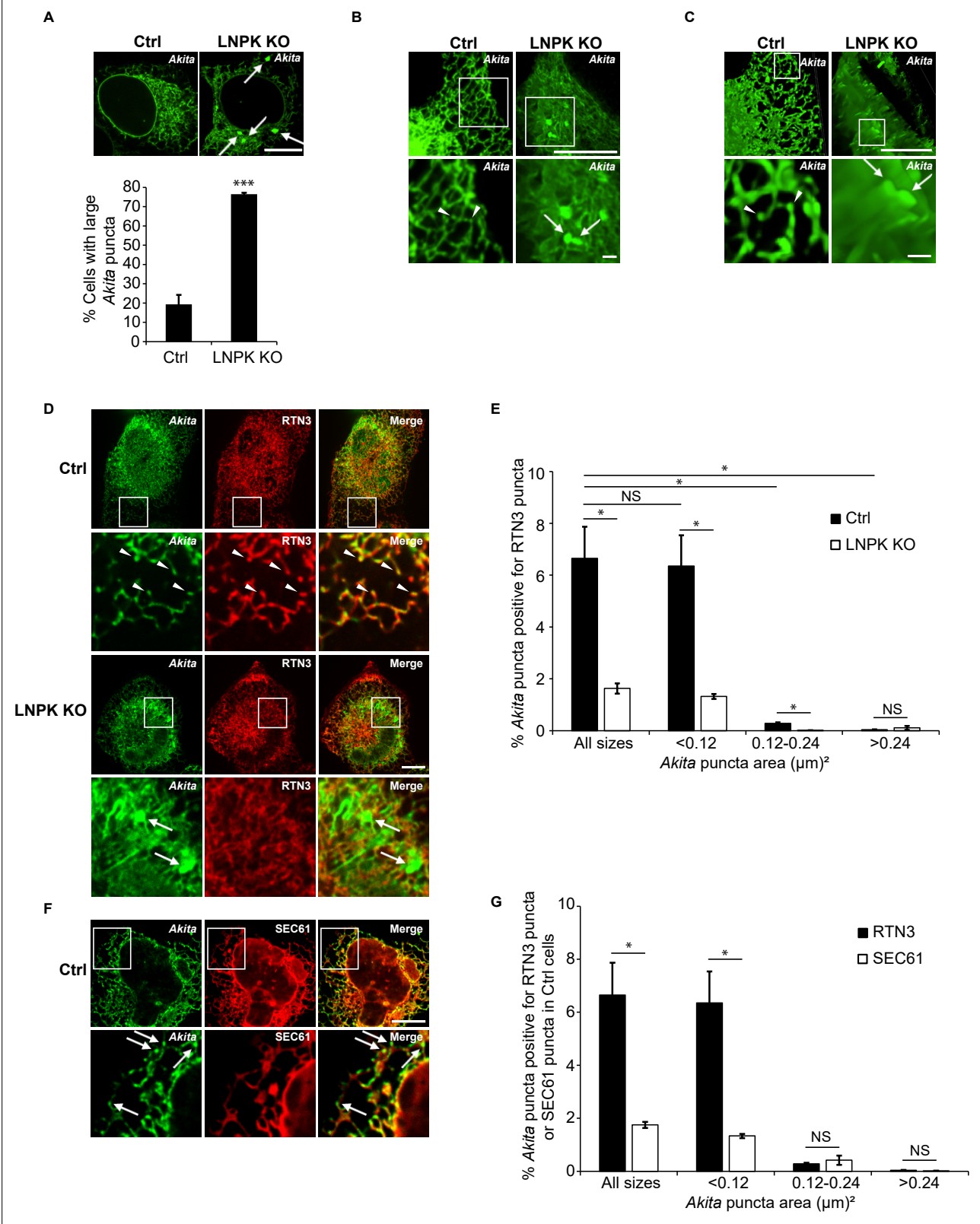

**Figure 5.** Large *Akita* puncta accumulate in LNPK KO cells. (**A**) Control (Ctrl) and LNPK KO cells were analyzed for the accumulation of large *Akita* puncta (≥0.5 µm²) by confocal microscopy (top). Arrows point to large puncta. Bar graph shows the % of cells with large *Akita*-sfGFP puncta (bottom). (**B**) Maximum intensity projections showing *Akita* puncta in Ctrl and LNPK KO cells used for the 3D reconstructions shown in *Videos 3 and 4*. (**C**) Frames from representative 3D reconstructions of Z stacks of a Ctrl cell (left, 27.5 s in *Video 3*) and an LNPK KO cell (right, 39 s in *Video 4*) are shown.

*Figure 5 continued on next page*

*Figure 5 continued*

Arrowheads mark small *Akita* puncta, while arrows mark the large puncta. Scale bar for top images in (**B**) and (**C**), 10 µm. The insets that are enlarged at the bottom, 1 µm. (**D**) Ctrl or LNPK KO cells stably expressing mCherry-RTN3 were examined for colocalizing *Akita*-sfGFP puncta. Arrowheads in the inset indicate *Akita* puncta colocalizing with RTN3 puncta (top). Arrows show large *Akita* puncta (≥0.5 µm²) in LNPK KO cells that do not colocalize with RTN3 (bottom). (**E**) Bar graph for the data shown in (**D**) of % *Akita*-sfGFP puncta of different sizes that colocalize with mCherry-RTN3 puncta. (**F**) Control cells expressing *Akita*-sfGFP and mCherry-SEC61β were analyzed by confocal microscopy. Arrows mark *Akita* puncta. (**G**) Bar graph for the Ctrl data shown in (**D**) and (**F**) of % *Akita*-sfGFP puncta of different sizes that colocalize with either mCherry-RTN3 puncta or mCherry-SEC61β puncta. To identify *Akita*, RTN3, or SEC61 puncta, the Yen threshold algorithm, which identifies punctate accentuations on the network, was used. We then identified colocalized puncta (*Akita*-RTN3, *Akita*-SEC61) using the Boolean image calculator in ImageJ. Colocalizing puncta were calculated as follows: *Akita* puncta colocalized with RTN3 puncta/total *Akita* puncta × 100%  or *Akita* puncta colocalized with SEC61 puncta/total *Akita* puncta × 100% . Scale bars in (**A**), (**D**), and (**F**), 10 µm. Error bars in (**A**), (**E**), and (**G**) represent SEM; n = 3 independent experiments. Approximately 30–40 cells were examined/ experiment for (**A**), and 15–30 cells/experiment for (**E**) and (**G**). NS: not significant (p≥0.05); *p<0.05, ***p<0.001, Student's unpaired *t*-test.

The online version of this article includes the following source data and figure supplement(s) for figure 5:

**Figure supplement 1.** Large *Akita* puncta accumulate when CLIMP63 is overexpressed.

**Figure supplement 2.** The delivery of *Akita* to lysosomes is reduced in LNPK KO cells or when CLIMP63 is overexpressed.

**Figure supplement 3.** hCOL1A1 puncta do not accumulate in LNPK KO cells.

**Figure supplement 4.** Proinsulin and *Akita* colocalize to the same large puncta.

**Figure supplement 5.** Large *Akita* puncta accumulate in INS1 cells in response to CLIMP63 overexpression.

**Figure supplement 5—source data 1.** Uncropped blots for *Figure 5—figure supplement 5D*.

**Figure supplement 6.** *Akita* does not colocalize with SEC24C and LC3B in LNPK KO cells.

**Figure supplement 7.** The velocity of small *Akita* puncta is unaltered in LNPK KO cells.

**Figure supplement 8.** Large *Akita* puncta rapidly recover after photobleaching in LNPK KO cells.

**Figure supplement 9.** Large *Akita* puncta rapidly recover after photobleaching in siRTN3 cells.

Together these studies have revealed that cargoes that are targeted for ER-phagy by the ER tubule receptor RTN3 form puncta that enlarge in sheet-like ER. A cargo that is targeted for ER-phagy by the ER sheets receptor, FAM134B, did not display this behavior.

*Akita* is known to be cytotoxic in insulin-secreting β cells (*Liu et al., 2007*). It exerts a dominant-negative phenotype by trapping wild-type proinsulin in high-molecular-weight oligomeric complexes that are retained in the ER, thereby disrupting insulin secretion (*Liu et al., 2007*; *Izumi et al., 2003*). If *Akita* behaves in a similar manner in U2OS cells, it should trap proinsulin in large puncta when both proteins are coexpressed. Interestingly, when *Akita*-sfGFP and proinsulin-FLAG were co-transfected in control and LNPK KO cells, 100%  of the large *Akita* puncta also contained proinsulin (*Figure 5— figure supplement 4A and B*). The same was also observed when G57S Pro-AVP-FLAG and Pro-AVP-HA (*Figure 5—figure supplement 4C and D*), or C28F POMC and POMC-Myc (*Figure 5—figure supplement 4E and F*) were co-transfected. Like *Akita*, G57S Pro-AVP-FLAG and C28F POMC exert dominant-negative phenotypes (*Kim et al., 2018*;

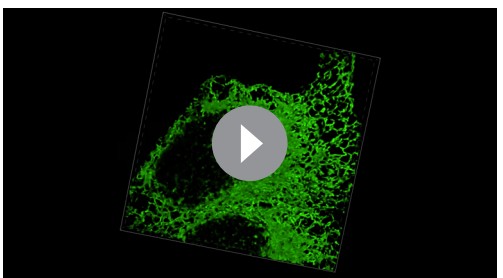

**Video 3.** Small *Akita* puncta are found in endoplasmic reticulum (ER) tubules. The movie shows a 3D reconstruction of 29 Z-slices, taken with 0.15 µm steps, of a single control cell with small *Akita*-sfGFP puncta. Images were deconvolved using Richardson–Lucy algorithm with automatic parameter setting in NIS Elements.

https://elifesciences.org/articles/71642/figures#video3

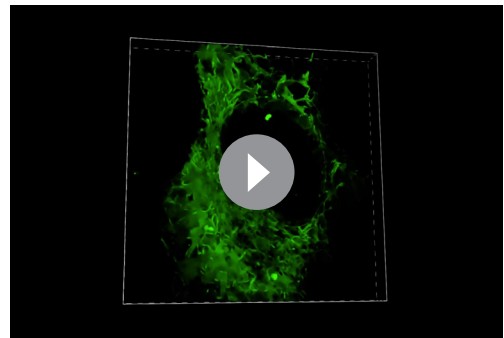

**Video 4.** Large *Akita* puncta accumulate in endoplasmic reticulum (ER) sheets in LNPK KO cells. The movie shows a 3D reconstruction of a Z stack of a single LNPK KO cell with large *Akita*-sfGFP puncta.

https://elifesciences.org/articles/71642/figures#video4

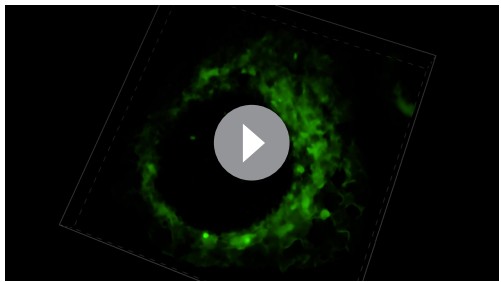

**Video 5.** Large *Akita* puncta accumulate in CLIMP63 overexpressing cells. The movie shows a 3D reconstruction of a Z stack of a control cell expressing *Akita*-sfGFP and mCherry-CLIMP63. Large *Akita* puncta can be seen in the endoplasmic reticulum (ER) sheets.
https://elifesciences.org/articles/71642/figures#video5

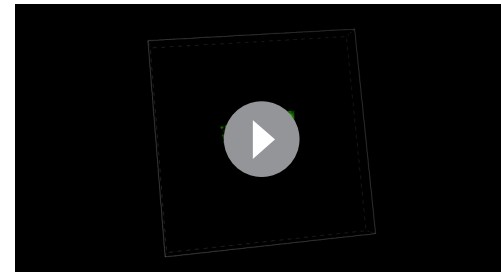

**Video 7.** A representative INS1 cell expressing *Akita*-sfGFP. The movie shows a 3D reconstruction of a Z stack of a single INS1 cell expressing *Akita*-sfGFP.
https://elifesciences.org/articles/71642/figures#video7

*Shi et al., 2017*). Together these findings suggest that *Akita*, G57S Pro-AVP, and C28F POMC behave similarly to the way they would in insulin-secreting or neuronal cells. As further evidence that our findings are physiologically relevant, we proliferated ER sheets by overexpressing CLIMP63 in insulin-secreting INS-1 832/13 rat β-cells (INS1 hereafter). Although ER tubules and sheets are difficult to visualize in INS1 cells (*Video 7*), the overexpression of CLIMP63 led to a significant increase in large *Akita* puncta (*Figure 5—figure supplement 5A–D*). Images of 3D reconstructions of Z stacks suggested that the large *Akita* puncta reside in sheet-like ER (see representative images in *Figure 5—figure supplement 5B*, *Video 8*).

## The association of *Akita* with RTN3 is required for the formation of ERPHS

To begin to understand why *Akita* puncta enlarge in LNPK KO cells, we analyzed the formation of ERPHS in the KO cells. Although the LNPK KO cells contained the same number of SEC24C puncta as control cells (*Figure 5—figure supplement 6A and B*), less *Akita* puncta colocalized with SEC24C (*Figure 5—figure supplement 6A and C*), and fewer cells contained *Akita* puncta that colocalized with LC3B (*Figure 5—figure supplement 6D and E*). The reduction in ERPHS is consistent with the decreased delivery of *Akita* to lysosomes in mutant cells (*Figure 5—figure supplement 2A and B*).

We reasoned that the decreased colocalization of *Akita* with SEC24C in LNPK KO cells might result from a failure of *Akita* to associate with RTN3. When we performed colocalization studies of *Akita* puncta with RTN3 puncta in control cells, only the small *Akita* puncta ($<0.12$ μm$^2$) associated with RTN3 puncta (*Figure 5D*, top, and *Figure 5E*). Colocalizing puncta were mostly detected at the tips and junctions of ER tubules (arrowheads, *Figure 5D*, top), and also observed when autophagosome formation was blocked (*Figure 5—figure supplement 7A*). Although the diffuse pool of *Akita*

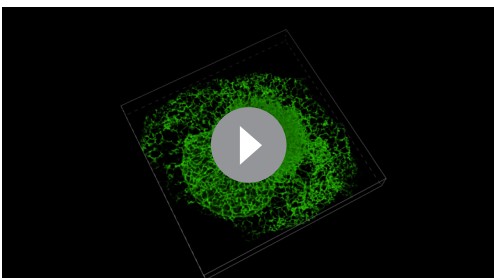

**Video 6.** Large *Akita* puncta decrease in number when CLIMP63 is cotransfected with RTN4. The movie shows a 3D reconstruction of a Z stack of a control cell expressing *Akita*-sfGFP, mCherry-CLIMP63, and mTurquiose2-RTN4. Small *Akita* puncta can be seen in endoplasmic reticulum (ER) tubules.
https://elifesciences.org/articles/71642/figures#video6

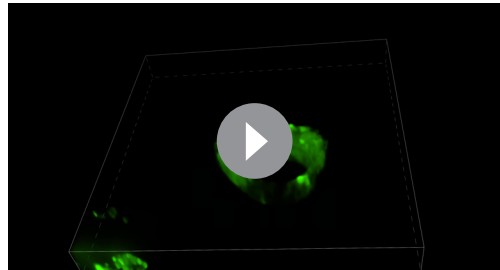

**Video 8.** The overexpression of CLIMP63 increases large *Akita* puncta in INS1 cells. The movie shows a 3D reconstruction of a Z stack of a single INS1 cell coexpressing *Akita*-sfGFP and mCherry-CLIMP63. Large *Akita* puncta can be seen in sheet-like endoplasmic reticulum (ER).
https://elifesciences.org/articles/71642/figures#video8

in the ER network colocalized with the pan ER marker SEC61 in the network, the SEC61 puncta that formed on the ER did not colocalize with *Akita* puncta (*Figure 5F and G*). Thus, the colocalization of small *Akita* puncta with RTN3 puncta was specific. In contrast to control cells, *Akita* puncta did not colocalize with RTN3 puncta in LNPK KO cells (*Figure 5D and E*, bottom). Large *Akita* puncta (arrows) were largely seen in sheet-like dense regions of the ER, while RTN3 was frequently seen at the cell periphery in the KO cells (*Figure 5D*, bottom). The inability of small *Akita* puncta to colocalize with RTN3 puncta in LNPK KO cells was not a consequence of decreased puncta movement in the network as the velocity of the small puncta was not significantly altered in mutant cells (*Figure 5—figure supplement 7B*). The large *Akita* puncta in LNPK KO cells also did not colocalize with SEC61 puncta, but were frequently seen in the ER lumen (*Figure 5—figure supplement 7C and D*). FRAP analysis of the large puncta revealed that *Akita*-sfGFP in the puncta rapidly exchanged with the *Akita*-sfGFP in the network of mutant cells (*Figure 5—figure supplement 8A–D*). The *Akita*-sfGFP in the large puncta also rapidly exchanged with the ER network pool in siRTN3-depleted cells (*Figure 5—figure supplement 9A–D*), indicating that *Akita* did not aggregate in the absence of RTN3. Together these findings imply that the failure of *Akita* to access RTN3 in LNPK KO cells is a consequence of its exclusion from tubules.

## Segregating *Akita* puncta into ER tubules prevents them from enlarging

Thus far our findings imply that *Akita* puncta that fail to access RTN3 puncta in tubules cannot undergo ER-phagy and instead enlarge in the sheets. If the decrease in three-way junctions and the consequential collapse of the tubular polygonal network in LNPK KO cells is what leads to the increase in large *Akita* puncta, restoration of the network should decrease the number of puncta. To address this possibility, we partially restored the network in LNPK KO cells by overexpressing the reticulon proteins, RTN3 or RTN4, and then quantified the accumulation of large *Akita* puncta. Overexpressed RTN3 increases the density of three-way ER junctions, while RTN4 drives ER tubulation (*Wang et al., 2016*; *Wu and Voeltz, 2021*). Unlike RTN3, RTN4 does not associate with LC3B or other Atg8 family members (*Grumati et al., 2017*). Consistent with previous studies showing that the loss of LNPK does not reduce the level of RTN4 (*Wang et al., 2016*), we found that RTN3 levels were also unaltered in the KO cells (*Figure 6A*). The sheet-like ER was partially converted to ER tubules in LNPK KO cells transfected with either RTN3 or RTN4 (*Figure 6B,C*). As the number of cells with sheet-like ER decreased, the percentage of cells with large *Akita* puncta decreased (*Figure 6C,D*), and small *Akita* puncta became more visible in the ER tubules (*Figure 6B*). The overexpression of RTN3 or RTN4 reduced the number of large *Akita* puncta equally as well, indicating that the reduction in large puncta was driven by the formation of tubules.

If the overexpression of RTN3 or RTN4 suppresses the accumulation of large *Akita* puncta in LNPK KO cells by driving ER-phagy in tubules, there should be an increase in the delivery of *Akita* to lysosomes in RTN3 or RTN4-overexpressing cells. When we transfected KO cells with *Akita*-sfGFP and either mTurquoise2-RTN3 or mTurquoise2-RTN4, we found that more *Akita* was delivered to lysosomes when sheet-like ER was converted to tubules (*Figure 6E and F*, *Figure 6—figure supplement 1*). Together these findings indicate that *Akita* can only be targeted to ER-phagy in tubules. In the absence of tubules, *Akita* puncta enlarge in sheets.

## Discussion

Here we analyzed three different dominant-interfering prohormones (*Akita*, G57S Pro-AVP, and C28F POMC) that are targeted to ER-phagy in ER tubules. In studying mutant proinsulin *Akita*, we found that small highly mobile puncta ($<0.12\ \mu m^2$ in size) associated with RTN3, LC3B, and SEC24C-SEC23A in the ER (*Figure 7*). We named these colocalizing puncta *ER-phagy sites* (ERPHS; *Figure 7*) as they accumulated when autophagosome formation was blocked. Although proinsulin colocalized with the ERES marker SEC24A, *Akita* did not. We conclude that ERPHS, which contain *Akita*, are distinct from the ERES that target proinsulin to the secretory pathway. When ER-phagy was blocked, the *Akita* puncta enlarged ($\geq 0.5\ \mu m^2$). The large puncta resided in sheet-like ER that did not colocalize with RTN3 or SEC24C, implying that SEC24C only sequesters *Akita* into autophagosomes on tubules (*Figure 7*). This observation is consistent with studies showing that COPII coat subunits reside on

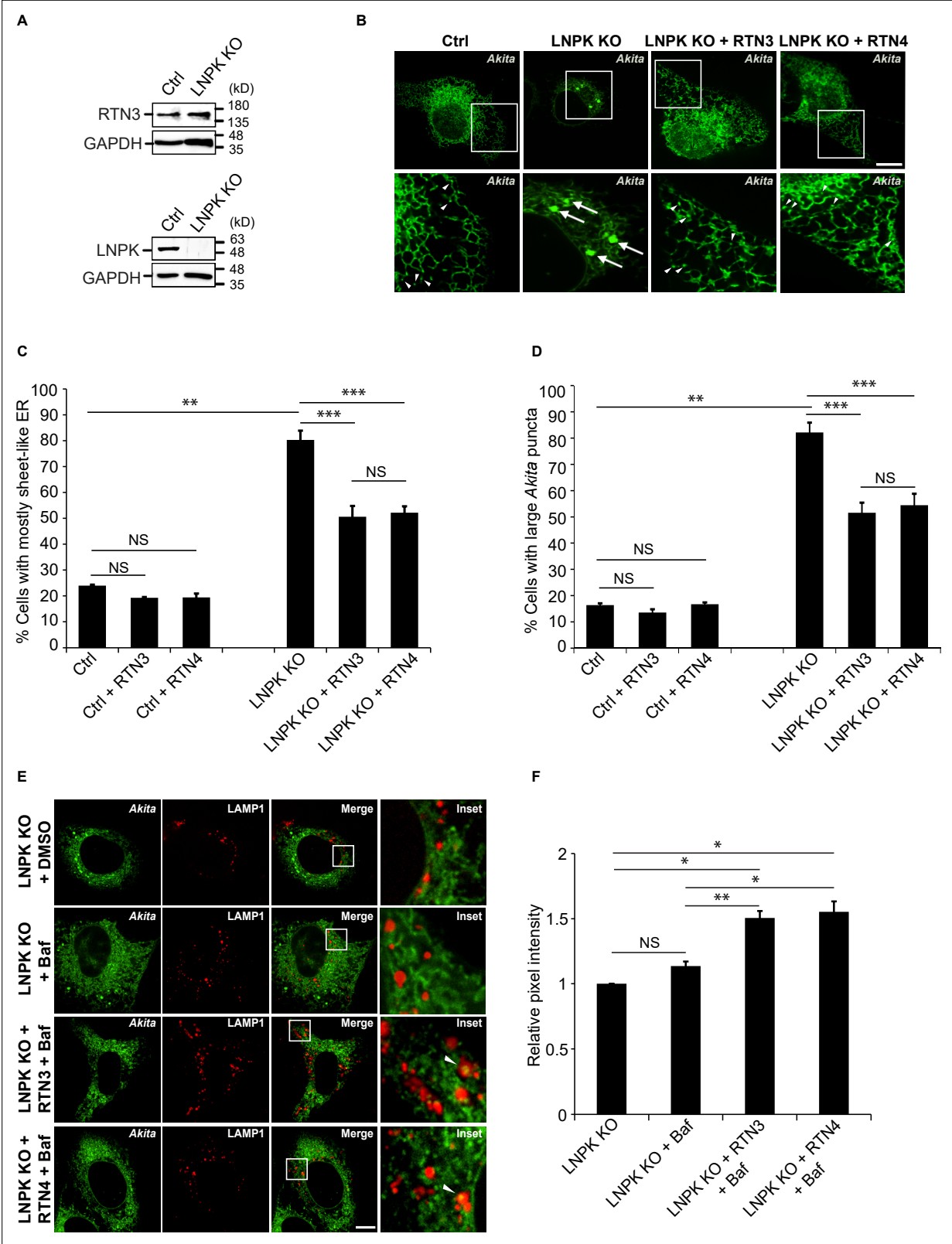

**Figure 6.** Restoration of the tubular endoplasmic reticulum (ER) network in LNPK KO cells reduces the accumulation of large *Akita* puncta. (**A**) Ctrl and LNPK KO cells were analyzed for the expression of endogenous RTN3 (top) or LNPK (bottom) by immunoblotting. Glyceraldehyde 3-phosphate dehydrogenase (GAPDH) was used as a loading control. (**B**) mCherry-RTN3 or mCherry-RTN4 was transfected in control and LNPK KO cells expressing *Akita*-sfGFP and analyzed by confocal microscopy. Arrowheads mark representative small *Akita* puncta present in the ER tubules. Arrows mark large

*Figure 6 continued on next page*

*Figure 6 continued*

*Akita* puncta (≥0.5 µm$^2$). (**C**) Bar graph showing the % of cells with mostly sheet-like ER for the data shown in (**B**). (**D**) Bar graph showing the % of cells with large *Akita*-sfGFP puncta for the data shown in (**B**). (**E**) mTurquoise2-RTN3 or mTurquoise2-RTN4 was transfected in control and LNPK KO cells expressing *Akita*-sfGFP and LAMP1-mCherry, and treated with Baf. (**F**) Quantitation of *Akita*-sfGFP in LAMP1-mCherry structures for the data shown in (**E**). The DMSO control for each condition was set to 1.0. Scale bar in (**B**) and (**E**), 10 µm. Error bars in (**C**), (**D**), and (**F**) represent SEM; n = 3 independent experiments. Approximately 30–40 cells were quantified in each experiment. NS: not significant (p≥0.05); *p<0.05, **p<0.01, ***p<0.001, Student's unpaired *t*-test.

The online version of this article includes the following figure supplement(s) for figure 6:

**Source data 1.** Uncropped blots for *Figure 6A*.

**Figure supplement 1.** The reticulons partially restored the tubular network in LNPK KO cells.

tubules in mammals and on highly curved membranes in yeast (*Okamoto et al., 2012*; *Hammond and Glick, 2000*). As the association of SEC24C with RTN3 is needed to target *Akita* to lysosomes, we conclude that SEC24C is an essential component of the RTN3-mediated ER-phagy pathway.

Previous studies have shown that ERAD-resistant high-molecular-weight oligomeric forms of *Akita* are cleared from the ER by RTN3 (*Cunningham et al., 2019*). Because these oligomers pelleted on sucrose density gradients, they were suggested to be aggregates (*Cunningham et al., 2019*). Our photobleach analysis of *Akita* puncta, showing that these structures behaved like liquid condensates and not aggregates, was surprising as condensates have not been observed within the ER before. This observation led us to discover that ER tubules play a critical role in restricting the size of condensates in the ER. Specifically, we found that the loss of ER tubules in LNPK KO cells resulted in a dramatic accumulation of large *Akita* puncta (≥0.5 µm$^2$; *Figure 7*). The puncta not only failed to be degraded in the absence of tubules, they continued to enlarge (*Figure 7*). The overexpression of RTN4, a protein that promotes tubule formation, partially restored the tubular network in LNPK KO cells and reduced the number of large puncta. RTN4 overexpression also increased the delivery of *Akita* to lysosomes. Because the loss of LNPK could indirectly affect ER shape in a number of ways, we used a second method to proliferate ER sheets. Consistent with the proposal that the *Akita* puncta enlarged in sheet-like ER, and not tubules, the proliferation of ER sheets by CLIMP63 overexpression increased the number of large *Akita* puncta and reduced the delivery of *Akita* to lysosomes. The co-transfection of CLIMP63 with RTN4 restored the tubular network in the CLIMP63-overexpressing cells and reduced the number of large *Akita* puncta. In total, these findings indicate that ER tubules, which have a small

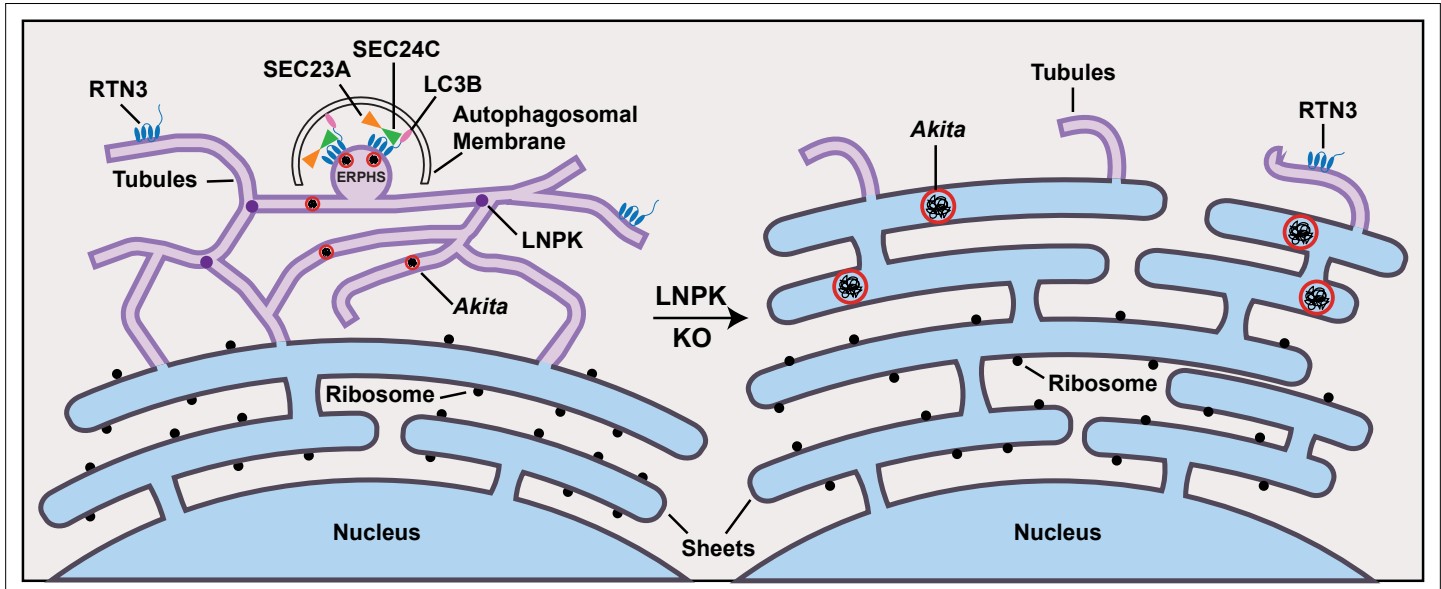

**Figure 7.** *Akita* puncta do not enlarge in the tubules. *ER-ph*agy sites (ERPHS) are formed when *Akita* puncta associate with RTN3 puncta, LC3B, and SEC24C on the tubules. The association of RTN3 with LC3B occurs in the absence of SEC24C. In LNPK KO cells, the tubular polygonal network collapses and large *Akita* puncta accumulate in sheet-like endoplasmic reticulum (ER). Large *Akita* puncta also accumulate in siSEC24C and siRTN3 cells.

outer radius of only ~44 nm (*Georgiades et al., 2017*), restrict the size of condensates. In a sheet, an *Akita* condensate can access the soluble milieu around its entire surface, while in a tubule, accessibility is limited to the ends facing the lumen. An *Akita* condensate is inaccessible to soluble oligomers in the area that is adjacent to the tubule wall. Therefore, although *Akita* condensates could expand along the length of tubules, their growth rate is limited by their reduced accessibility. Interestingly, other RTN3-SEC24C substrates (G57S Pro-AVP and C28F POMC) behaved like *Akita*, while a cargo that is targeted for lysosomal degradation in the sheets (hCOL1A1) did not. Together these findings imply that misfolded proteins, which form fluid condensates, are largely disposed of by ER-phagy in tubules.

Unlike aggregates, which are biologically inert, condensates have the ability to rapidly exchange their contents with the surrounding environment (*Banani et al., 2017*; *Noda et al., 2020*). Our FRAP studies showing that *Akita* in puncta rapidly exchanged with *Akita* in the ER provide a possible explanation for how *Akita* can trap secretory cargo and disrupt the local environment of the ER. Consistent with our findings, a comparison of the toxicity of liquid and aggregated states of α-synuclein has suggested that soluble oligomers of mutant α-synuclein, and not aggregated forms, are the toxic species in Parkinson disease (*Winner et al., 2011*). Similarly, in Alzheimer's disease, soluble Aβ oligomers and not amyloid fibrils have been shown to be neurotoxic (*Walsh et al., 2002*). As ER-phagy

**Table 1.** List of plasmids used in this study.

| Identifier | Plasmid | Source | Catalog no. |
|---|---|---|---|
| SFNB 2454 | mCherry-hLC3B-pcDNA3.1 | Addgene | #40827 |
| SFNB 2453 | pC1-EYFP-SEC24C | Addgene | #66608 |
| SFNB 2501 | pCDNA3.1-LAMP1-mCherry | Addgene | #45147 |
| SFNB 2498 | pLVX-Puro-mRuby-SEC23A | Addgene | #36158 |
| SFNB 2497 | pHAGE2-mCherry-RTN4a | Addgene | #86683 |
| SFNB 2502 | pC1-mCherry-SEC61beta | Addgene | #49155 |
| SFNB 2508 | pcDNA 3.1- EGFP-hCOL1A1 | Addgene | #140110 |
| SFNB 2535 | pmCherry-N1-CLIMP63 | Addgene | #136293 |
| SFNB 2462 | Proinsulin-sfGFP (also called pTarget-hProCpep SfGFP) | *Haataja et al., 2013* | N/A |
| SFNB 2468 | *Akita*-sfGFP (also called pTarget- hProC(A7)Y-Cpep SfGFP) | *Haataja et al., 2013* | N/A |
| SFNB 2455 | pHAGE-mCherry-RTN3L | *An et al., 2019* | N/A |
| SFNB 2456 | pHAGE-FAM134B-mTurquoise | *An et al., 2019* | N/A |
| SFNB 2467 | pC1-mAaus0.5-SEC24A | This study | N/A |
| SFNB 2477 | pC1-mCherry-SEC24A | This study | N/A |
| SFNB 2471 | pC1-mCherry-SEC24C | This study | N/A |
| SFNB 2457 | SEC13-GFP | *Hammond and Glick, 2000* | N/A |
| SFNB 2458 | EGFP-SEC16A | *Watson et al., 2006* | N/A |
| SFNB 2489 | pRc/RSV-G57S Pro-AVP-FLAG | *Shi et al., 2017* | N/A |
| SFNB 2491 | pcDNA3.1-C28F POMC-FLAG | *Kim et al., 2018* | N/A |
| SFNB 2513 | pC1-mTurquoise2-RTN3 | This study | N/A |
| SFNB 2514 | pC1-mTurquoise2-RTN4 | This study | N/A |
| SFNB 2536 | pC1-mTurquoise2-CLIMP63 | This study | N/A |
| SFNB 2523 | pC1-Proinsulin-FLAG | This study | N/A |
| SFNB 2517 | pcDNA3.1-POMC-WT-myc | *Kim et al., 2018* | N/A |
| SFNB 2518 | pRc/RSV- Pro-AVP-WT-HA | *Shi et al., 2017* | N/A |

**Table 2.** List of antibodies used in this study.

| Name | Source | Catalog no. | Dilution used |
|---|---|---|---|
| Rabbit anti-SEC24A | Gift from Randy Schekman | N/A | 1:1000 |
| Rabbit anti-SEC24B | Gift from Randy Schekman | N/A | 1:400 |
| Rabbit anti-SEC24C | Gift from Randy Schekman | N/A | 1:1000 |
| Rabbit anti-SEC24D | Gift from Randy Schekman | N/A | 1:1000 |
| Mouse anti RTN3 (F-6) | Santacruz Biotechnology | sc-374599 | 1:1000 |
| Rabbit anti-LNPK | *Chen et al., 2015* | N/A | 1:1000 |
| Rabbit anti FAM134B | Proteintech Group, Rosemont, IL | 21537-1-AP; 00014408 | 1:1000 |
| Mouse GAPDH loading control monoclonal antibody (GA1R) | Thermo Fisher Scientific, Waltham, MA | MA5-15738 | 1:5000 |
| Anti-mouse IgG (H + L), HRP conjugate | Promega | W4021; 0000421603 | 1:10,000 |
| Rabbit IgG HRP linked whole Ab | Millipore Sigma | GENA934 | 1:10,000 |
| Mouse monoclonal anti-FLAG M2 antibody | Millipore Sigma | F3165; SLB45142 | 1:1000 |
| HA-Tag (C29F4) rabbit mAb | Cell Signaling | 3724T | 1:1000 |
| Myc-Tag (71D10) rabbit mAb | Cell Signaling | 2278T | 1:1000 |
| Anti-GFP mouse $IgG_1 \kappa$ (clones 7.1 and 13.1) | Roche | 11814460001 | 1:3000 |
| Rabbit anti-SEC24C (for immunofluorescence and western blots) | Bethyl | A304-759A | 1:1000 |
| Rabbit anti-Nogo (RTN4) A + B antibody | Abcam | ab47085 | 1:1000 |
| Rabbit anti-LC3 pAb | MBL | PM036 | 1:1000 |
| Goat anti-mouse IgG (H + L) cross-adsorbed secondary antibody, Alexa Fluor 488 or 594 | Thermo Fisher Scientific | A-11001 A-11005 | 1:1000 |
| Goat anti-rabbit IgG (H + L) highly cross-adsorbed secondary antibody, Alexa Fluor 488 or 594 | Thermo Fisher Scientific | A-11008 A-11037 | 1:1000 |

is induced slowly (*Cui et al., 2019*), misfolded cargoes can accumulate in the ER before they are removed. The ability of tubules to restrict condensate size averts potentially toxic condensates from expanding in the ER while they are waiting to undergo ER-phagy. In total, our findings suggest that drugs that reduce the size of *Akita* puncta in the ER could be of therapeutic value in the intervention of MIDY. We also speculate that the observations we have made here can be extended to a variety of other disorders that are associated with the accumulation of dominant-interfering, ERAD-resistant proteins in the ER.

## Materials and methods
### Plasmids and antibodies
Plasmids used in this study are listed in *Table 1*. *Akita*-sfGFP (*Akita* fused to superfolder GFP) was a gift from Peter Arvan (University of Michigan Medical School). mAaus0.5-SEC24A, mCherry-SEC24A, and mCherry-SEC24C were generated by PCR amplification using pCS-6x-Myc-SEC24A or pC1-EYFP-SEC24C, respectively, as a template and then the PCR fragment was cloned into pC1 mammalian expression vector (Promega, Madison, WI) with a N-terminal mAaus0.5 tag or an N-terminal mCherry tag. The FP tag mAaus0.5 is a monomeric variant of the ultra-bright FP AausFP1 (*Lambert et al., 2020*) (generated by structure-guided directed evolution). All clones were generated using the Gibson assembly method (*Gibson et al., 2009*). Antibodies used in this study are listed in *Table 2*.

## Cell culture

U20S cells were purchased from the American Type Culture Collection (Manassas, VA) and authenticated by STR analysis. The cells were tested for the absence of mycoplasma using the PlasmoTest -Mycoplasma detection kit (InvivoGen; rep-pt1; San Diego, CA).

The LNPK KO cells were obtained from Dr. Tom Rapoport's lab, and the INS-1 832/13 cells were obtained from Dr. Maike Sander. The U2OS cells were grown at 37 °C with 5 % $CO_2$ in DMEM medium (Gibco; 11965-092; Paisley, UK) supplemented with 10% fetal bovine serum (Gibco; 16000044) and 100 U/ml Penicillin-Streptomycin-Glutamine (Gibco; 10378016). INS-1 832/13 cells were cultured at 37 °C with 5% $CO_2$ in RPMI-1640 medium containing 11.1 mM glucose (Gibco; 11875093) and supplemented with 10% fetal bovine serum, 10 mM HEPES pH 7.4 (Gibco; 15630080), 1 mM sodium pyruvate (Gibco; 11360070), 50 µM β-mercaptoethanol (Gibco; 31350010) and 100 U/ml Penicillin-Streptomycin-Glutamine. The extracellular medium was collected 48 hr after transfection, and insulin secretion was monitored as described in the legend to *Figure 5—figure supplement 5D*.

## Generation of stable cell lines

To prepare lentivirus used for the generation of stable cell lines, HEK293T cells were transiently transfected with lentiviral packaging vectors and pHAGE-mCherry-RTN3L using TransIT-LT1 (Mirus; MIR2306; Madison, WI). The cells were harvested 2 days post-transfection and the supernatant was passed through a 0.45 µm syringe filter unit (Millipore; HAWP04700; Burlington, MA) and frozen at –80 °C. Subsequently, control and LNPK KO U2OS cells were transduced with lentivirus and selected with 1.5–2 µg/ml of puromycin (Gibco; A1113803).

## Transfection

For transient expression, U2OS cells were transfected with plasmid DNA using TransIT-LT1 (Mirus; MIR2306) or TransFast (Promega; E2431) transfection reagent according to the manufacturer's instructions and imaged 24–48 hr after transfection. For INS-1 832/13 cells, Lipofectamine-2000 (Thermo Fisher; 11668019) transfection reagent was used, and the cells were imaged 48 hr after transfection.

**Table 3.** List of siRNAs used in this study.

| Name | Target gene | Source | Catalog no. | Target sequence |
|---|---|---|---|---|
| siCtrl | Control | Dharmacon (siGenome Non-Targeting siRNA Pool) | D-001206-13-05 | |
| siSEC24A | SEC24A | Dharmacon (siGenome-SMARTpool) | M-024405-01-0005 | CCAAGAAGGUAUUACAUCA CAAAUGCACGUCUAGAUGA GGAAACUUCUUUGUUAGGU GGUUGUAUUUCUCGGUAUU |
| siSEC24B | SEC24B | Dharmacon (siGenome-SMARTpool) | M-008299-01-0006 | GGGAAAGGCUGUGACAAUA GACCAGAAGUUCAGAAUUC CAGGGUGCAUCUAUUAUUA CCAGAUUCAUUUCGGUGUA |
| siSEC24C | SEC24C | Dharmacon (siGenome-SMARTpool) | M-008467-01-0007 | GCAAACGUGUGGAUGCUUA CAGGGAAGCUCUUUCUAUU UGGCUGAUCUAUAUCGAAA CUGUAUAUGAUUCGGUAUU |
| siSEC24D | SEC24D | Dharmacon (siGenome-SMARTpool) | M-008493-01-0008 | GAGGAACCCUUUACAAAUA GACCAGAGAUCUCAACUGA GUACAUGAAUUGCUUGUUG GGUAAAUCACGGCGAGAGU |
| siRTN3 | RTN3 | Sigma-Aldrich | N/A | UCAGUGUCAUCAGUGUGGUUUCUUAdTdT |
| siFAM134B | FAM134B | Sigma-Aldrich | N/A | CAAAGATGACAGTGAATTAdTdT |

## Small interfering RNA knockdowns

The RNAi oligonucleotides listed in *Table 3* were transfected into cells using HiPerFect transfection reagent (Qiagen; 301707; Germantown, MD) according to the manufacturer's instructions. For the knockdown of SEC24 isoforms, 10 nM siRNAs were mixed with 6 μl of HiPerFect transfection reagent in 100 μl of serum-free DMEM and added to freshly plated cells. The cells were then analyzed 72 hr after siRNA transfection. The isoform specificity of the SEC24 siRNAs is shown in *Figure 2B* and *Figure 2—figure supplement 1B*. For RTN3 knock downs, 10–20 nM siRNA was mixed with 6–12 μl of HiPerFect transfection reagent in 100 μl of serum-free DMEM and cells were transfected as described above. For the siFAM134B knock downs, 80 nM siRNA was used. The siRTN3 and FAM134B oligos used in this study were previously tested for off-target effects (*Cunningham et al., 2019*).

## Western blotting

To prepare cell lysates, harvested cells were resuspended in RIPA buffer (Cell Signaling Technology; 9806) containing 1× protease inhibitor (Roche; 11697498001; Basel, Switzerland) and incubated on ice for 30 min with intermittent vortexing. Afterwards, the lysate was heated at 65 °C for 10 min in Laemmli buffer on a Thermal mixer-shaker with continuous shaking and then fractionated on an 8% SDS polyacrylamide gel for immunoblotting. To measure the endogenous level of RTN3 in control and LNPK KO cells, cells were harvested at 80% confluency, lysed as described above, and immunoblotted with anti-RTN3 antibody. To confirm the absence of LNPK in the LNPK mutant, the cells were harvested at 80% confluency, lysed as described above, and immunoblotted using anti-LNPK antibody. To normalize the total protein levels in lysates, the samples were immunoblotted with anti-GAPDH antibody.

## Fluorescence microscopy

For the live-cell imaging experiments, the cells were cultured on glass-bottom dishes (Cellvis; D35-20-1.5-N; Mountain View, CA) and imaged on two different spinning disk confocal microscopes. Either a Yokogawa spinning disk confocal microscope (Observer Z1; Carl Zeiss, Oberkochen, Germany), equipped with an electron-multiplying CCD camera (QuantEM 512SC; Photometrics), or a Nikon Ti2-E microscope was used. The Nikon microscope employed a Yokogawa X1 spinning disk confocal system with a 100× Apo TIRF 1.49 NA objective, MLC400B 4-line (405 nm, 488 nm, 561 nm, and 647 nm) dual-fiber laser combiner (Agilent), Prime 95B back-thinned sCMOS camera (Teledyne Photometrics), piezo Z-stage (Mad City Labs), and running NIS Elements software. Cells were imaged while in a stage top environmental chamber (Tokai Hit). To separate the emission of individual fluors, band-pass emission filters were used for each channel (450/50, 525/36, 605/52, and 705/72). To optimally acquire 3D images, Z-planes were collected with 150 nm Z-steps. The resulting images were then deconvolved using the Richardson–Lucy algorithm with automatic detection and setting of the SNR and optimal iteration number.

For the experiments where ER-phagy was induced with Torin, cells were treated for 3.5 hr with 250 nM Torin 2 (Sigma-Aldrich; SML1224; St. Louis, MO) that was solubilized in DMSO. DMSO-treated cells were used as a control. To inhibit autophagy, cells were treated with 2 μM of MRT68921 (Tocris; 5780; Bristol, UK) for 3.5 hr or 6 hr before imaging.

For immunofluorescence experiments, the cells were cultured on glass-bottom dishes and fixed in 4% paraformaldehyde (PFA) for 15 min at room temperature (RT). The fixed cells were washed with phosphate buffered saline (PBS, pH 7.4), incubated in 50 mM NH$_4$Cl for 15 min at RT, and then washed again with PBS. Subsequently, cells that were imaged for *Akita*, Pro-AVP, or POMC were incubated in permeabilization buffer (0.1% Triton-X-100 in PBS) for 10 min at RT, washed with PBS, and incubated in blocking buffer (1% BSA, 0.05% Tween-20 in PBS) for 1 hr at RT before they were washed with PBS. Samples were then incubated with primary antibody for 12–16 hr at 4 °C, washed with washing buffer (PBS containing 0.05% Tween-20), and incubated for 1 hr with Alexa Fluor 488/594 labeled secondary antibody at RT. Before imaging, the samples were washed in washing buffer and rinsed with PBS.

## Measurement of *Akita* delivery to lysosomes

For analyzing *Akita* inside LAMP1 structures, cells expressing *Akita*-sfGFP and LAMP1-mCherry were treated with 100 nM bafilomycin A1 (Sigma-Aldrich; B1793) for 1.5–4 hr before they were imaged. To inhibit autophagy, the cells were treated with MRT68921 for 6 hr and bafilomycin A1 was added

during the last 90 min of the incubation. Samples were fixed with 4% PFA for 6 min at RT, washed with PBS, incubated in 50 mM $NH_4Cl$ for 8 min at RT, washed with PBS, and imaged as described in the previous section. Relative pixel intensity was plotted. The relative pixel intensity for each condition is the mean intensity of *Akita*-sfGFP in the pixels that overlap with LAMP1. The data was normalized to the mean intensity obtained in the siCtrl condition. Approximately 1–2% of the total *Akita* in the cell is delivered to the lysosome via ER-phagy. This number was estimated by dividing the area of *Akita* that overlaps with LAMP1 by the total *Akita* in the cell.

### Image analysis

To quantitate the colocalization of two proteins, we employed an object overlap method in ImageJ v2.0. Briefly, an individual cell was cropped using an enclosing rectangular region of interest (ROI), and the resulting two-channel image was split into two images, one for each color channel. Each image was then subjected to uniform and unbiased auto-thresholding using the Yen algorithm to produce binary black-and-white images with black objects displaying the labeling pattern for each protein. This particular thresholding algorithm captured isolated spots as well as punctate accentuations on larger objects such as tubules, making it ideal for quantitation of ERPHS proteins. The Image Calculator was then used to create an intersection image of the two binary images using the Boolean AND operator. This binary image represents all labeled objects that are common to both proteins. The Analyze Particles command was then used to determine the total object count of all three binary images individually. The object count of the intersection image was then divided by the object count of one of the source images to generate percent colocalization. For example, for two proteins (A and B), the percent colocalization of A with B = [count of A AND B]/[count of A] × 100 %.

To measure the intensity of a protein localized to specific sites, individual channel component images were created from a two-color image as described above. One of the component images was used as the site of interest, for example, in *Figure 2A*, LAMP1 was used to mark lysosomes and autolysosomes. The site image was then subjected to auto-thresholding to produce a binary image as described above. The Analyze Particles command was then used to obtain a mask of the objects, and the Create Selection command was used to capture all the objects as a single ROI, which was then moved into the ROI Manager. A rectangular background ROI was drawn by hand in a dark extracellular region and moved into the ROI Manager. The two ROIs were then applied consecutively to the non-binary component image (e.g., *Akita* in *Figure 2A*) and the Measure command was used to obtain mean intensities. The pixel intensity (in arbitrary units) of the measured protein at the sites of interest = [mean intensity of sites ROI] – [mean intensity of background ROI]. This value collected from each condition was then divided by the mean pixel intensity from the control condition to produce relative pixel intensity for each condition.

For determination of the size of puncta, the binary black-and-white image was generated as described above and the Analyze Particles command was used to create a mask of the images (size criteria of <0.12, 0.12–0.24, >0.24, or ≥0.5 $\mu^2$ and circularity criteria of 0.5–1). The cells with puncta of size ≥0.5 μm$^2$ were quantified as the cells with large puncta. To determine the size-based colocalization of *Akita* with different proteins, the masks of *Akita* puncta of indicated size ranges were generated as described above and Image Calculator was then used to create an intersection image of the two proteins using the Boolean AND operator. Colocalization was then determined as described above.

To calculate the mean size of *Akita* puncta per cell, the Yen thresholding algorithm was first used to identify puncta of all sizes. Then, the number and area of *Akita* puncta were calculated using the Analyze Particles command in ImageJ. An average *Akita* puncta size for each cell was calculated as follows: total area of all *Akita* puncta/total number of *Akita* puncta in the cell.

To quantify the percent of cells with multiple *Akita* puncta colocalizing with either SEC24C, SEC24A, or LC3B, the images were analyzed for colocalization as described above using ImageJ, and the cells showing two or more colocalizing puncta were divided by the total number of cells analyzed and then multiplied by 100% .

To calculate the percentage of cells with mostly sheet-like ER, cells that largely contained sheets were blindly scored as positive for the appearance of sheets by two independent observers and divided by the total number of cells scored. The data from the two independent observers was then averaged for each experiment.

Immunofluorescence was also performed to measure the transient and stable expression levels of fluorescently tagged SEC24C, RTN3, LC3B, and RTN4, relative to the endogenous levels of these proteins. For these experiments, fixed cells were permeabilized with 0.1% saponin in PBS for 10 min at RT and then incubated in blocking buffer (1% BSA, 0.1% saponin in PBS) for 1 hr. Subsequently, the samples were incubated for 12–16 hr at 4 °C with either anti-SEC24C, RTN3, LC3B, or RTN4 antibody prepared in blocking buffer, and then treated with secondary antibody. The expression of each fluorescently tagged protein was estimated in the following way. Transfected versus non-transfected cells were identified in the color channel of the fluorescently tagged protein. Then, using the immunofluorescence image channel, an ROI was drawn around the entire cytoplasmic area of each cell to be analyzed, and the ROI was used to obtain the mean intensity. Approximately 10–15 transfected and non-transfected cells were examined for each marker protein in three separate experiments. The mean intensity of each transfected cell was divided by the mean of all non-transfected cells for each marker to obtain the fold overexpression for each transfected cell. Please note that highly overexpressing transient or stably expressing fluorescent cells were excluded in our colocalization studies and therefore also excluded from these estimates. For the markers that were introduced into cells by transient transfection (see list above), no more than twofold average overexpression was observed. For the mCherry-RTN3 stable cell lines, approximately 4–5.5-fold overexpression was observed. With the exception of *Figure 1C and D*, all experiments shown in the article with mCherry-RTN3 were performed with stable cell lines. In addition, all colocalization experiments with the mCherry-RTN3 stable cell lines were confirmed with transiently transfected mCherry-RTN3 cells. Identical results were obtained with stable versus transfected cells.

### Time-lapse imaging and analysis of the velocity of *Akita* puncta

U2OS cells were transfected with *Akita*-sfGFP and seeded on a glass-bottom culture dish 24 hr after transfection. Before the cells were imaged, the culture medium was changed to OPTI-MEM medium. Time-course images were captured without a delay using a Nikon spinning disc confocal microscope described above. The motility and velocity of the *Akita* puncta were tracked and measured using MTrackJ software. The velocity of the puncta was calculated as calibrated total (not net) linear distance of travel per unit time of the recording. The movies were generated using ImageJ software.

### Fluorescence recovery after photobleaching (FRAP) assay

U2OS cells transfected with *Akita*-sfGFP were imaged using a Nikon C2 confocal microscope attached to a Ti2-E with a Plan Apo 100× 1.45 NA objective. The C2 used a four-line (405 nm, 488 nm, 561 nm, and 640 nm) LUN-4 laser engine and a DUV-B tunable emission dual-GaAsP detector. To increase imaging speed to two frames per second, images were taken at 256 × 256 pixels with a zoom set to create 0.15 m pixels. The pinhole was fully opened to allow for optimal detection of emitted photons. For scanning, the 488 nm laser was set to 1%, and for bleaching the laser intensity was set to 100%. Six prebleached images were acquired without a delay; postbleaching images were taken continuously for 30 s and with 2 s intervals for the subsequent 2 min. The Time Measurement tool within NIS Elements was used to quantify the intensity of the bleached ROIs as well as the entire cell after the background intensity was subtracted from the images. FRAP quantitation was performed according to the method of Pfair and Misteli (*Phair and Misteli, 2000*).

### Statistical analysis

p-Values were calculated using the Student's *t*-test. Three or more independent experiments were used to report the statistical significance, which is presented as a mean value. The error bars in the figures represent SEM unless otherwise indicated in the legends.

## Acknowledgements

We thank Drs. Peter Arvan, Wade Harper, Ling Qi, Tom Rapoport, and Randy Schekman for plasmids, cells, antibodies, and Drs. Maike Sander and Han Zhu for advice and the INS1 cells used in this study. We also thank Dr. Fulvio Reggiori for a critical reading of the manuscript, and Drs. James Shorter and Michael Rosen for advice during the preparation of this manuscript. We acknowledge Andrea Lougheed for preparing the model in *Figure 7* and the Nikon Imaging Center at UC San Diego for help with imaging.

# Additional information

## Funding

| Funder | Grant reference number | Author |
|---|---|---|
| National Institute of General Medical Sciences | 5R35GM131681 | Susan Ferro-Novick |
| National Institute of Neurological Disorders and Stroke | RO1NS117440 | Susan Ferro-Novick |
| National Institute of Diabetes and Digestive and Kidney Diseases | R01DK068471 | Matthew Wortham |
| National Institute of General Medical Sciences | 2R15GM106323 | Jesse C Hay |
| National Institute of General Medical Sciences | R01GM109984 | Nathan C Shaner |
| National Institute of General Medical Sciences | R01GM121944 | Nathan C Shaner |
| National Institute of Neurological Disorders and Stroke | U01NS099709 | Nathan C Shaner |
| National Eye Institute | R21EY030716 | Nathan C Shaner |
| National Institute of Neurological Disorders and Stroke | U01NS113294 | Nathan C Shaner |
| National Science Foundation | 1707352 | Nathan C Shaner |
| The Pathways in Biological Science Graduate Training Program | | Christina R Liem |

The funders had no role in study design, data collection and interpretation, or the decision to submit the work for publication.

## Author contributions

Smriti Parashar, Conceptualization, Data curation, Formal analysis, Investigation, Methodology, Project administration, Writing – original draft, Writing – review and editing; Ravi Chidambaram, Shuliang Chen, Gerard G Lambert, Data curation, Formal analysis, Investigation, Methodology; Christina R Liem, Methodology, Resources; Eric Griffis, Data curation, Formal analysis, Investigation, Methodology, Resources; Nathan C Shaner, Data curation, Formal analysis, Funding acquisition; Matthew Wortham, Methodology; Jesse C Hay, Conceptualization, Data curation, Formal analysis, Funding acquisition, Investigation, Methodology, Supervision, Writing – original draft, Writing – review and editing; Susan Ferro-Novick, Conceptualization, Data curation, Formal analysis, Funding acquisition, Investigation, Methodology, Project administration, Supervision, Writing – original draft, Writing – review and editing

## Author ORCIDs

Susan Ferro-Novick http://orcid.org/0000-0001-8714-7352

## Decision letter and Author response

Decision letter https://doi.org/10.7554/eLife.71642.sa1
Author response https://doi.org/10.7554/eLife.71642.sa2

## Additional files

### Supplementary files
• Transparent reporting form

### Data availability
All data generated or analyzed during this study are included in the manuscript and supporting files.

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
