## [Decision Letter]

[Editors’ note: the authors submitted for reconsideration following the decision after peer review. What follows is the decision letter after the first round of review.]

Thank you for submitting your work entitled "Endoplasmic reticulum tubules limit the size of misfolded protein condensates" for consideration by *eLife*. Your article has been reviewed by 3 peer reviewers, one of whom is a member of our Board of Reviewing Editors, and the evaluation has been overseen by a Senior Editor. The reviewers have opted to remain anonymous.

We are sorry to say that, after consultation with the reviewers, we have decided that your work will not be considered further for publication by *eLife*. Although we judge the work to be of considerable merit, each of the three reviewers had comments that we believe would require more than 2 months to address. The detailed comments of the reviewers follow this comment but in summary: Reviewer #1 had a significant concern (No.3); reviewer #2 had a significant concern (No.1 and 2); reviewer #3 had general concerns how the ectopic overexpression of several key proteins and the use of cell lines that do not necessarily relate to the physiological context of the Akita mutant and the normal process of insulin secretion. Should you be in a position to address these concerns, we would welcome a new submission of this work which we will endeavor to have reviewed by the same referees.

*Reviewer #1:*

Parashar et al. present interesting results on the organization of a specialized zone of the endoplasmic reticulum, ER tubules, that appear to organize the autophagic engulfment and diversion to the lysosome of a mutant, misfolded form of proinsulin, the Akita mutant protein. This specialized zone correlates to the localization of an ER membrane protein, RTN3, known to promote membrane tubulation and also to cytoplasmic subunits, SEC24C and SEC23A, of a coat protein complex that participates in sorting of membrane proteins destined to traffic the ER. In contrast to what had been assumed of aggregated forms of the Akita mutant proinsulin, the authors find that puncta of the accumulated protein remain in a fluid state, characteristic of protein condensates rather than of a more rigid protein aggregate.

Parashar et al. have investigated the localization and fluid properties of mutant Akita that accumulates in the ER and is subject to a process of turnover in the lysosome termed ER-phagy. The work relies on live cell imaging of transfected forms of fluorescnety tagged forms of Akita, of wild type prolinsulin, collagen, RTN3, Sec24C and mutant forms of other prohormones. The broad conclusions are consistent with and extend previous observations on the role of RTN3 in formation of ER tubules in relation to the ER-phagy of other misfolded proteins subject to lysosomal degradation.

The novel conclusions the authors make are that the Akita mutant protein is in a fluid protein condensate state rather than as an aggregate and that ER-phagy operates selectively on tubular membrane marked by their content of RTN3 in the membrane and SEC24C on the cytoplasmic face of the tubular membrane.

I have several concerns that could be addressed by additional experimental work and a few small issues with the text.

1. All of the conclusions of this work rely on visual inspection of fluorescently-tagged over-expressed proteins. At the very least, this should be expressed as a limitation of the analysis but it would help support the strength of the conclusions if at least some of the work was repeated by another approach. One such additional approach would be to use immunoprecipitation of RTN3 to detect a a possibly selective association with SEC24C under conditions that promote ER-phagy in non-transfected cells. If the endogenous level of these two proteins is too low, the authors might consider a proximity labeling approach with APEX-tagged Akita, RTN3 or SEC24C.

2. How is the Akita protein retained in proximity to RTN3 in the membrane? Is there a direct interaction or is the Akita condensate anchored by some other means to the putative tubular regions of the ER?

3. The claim of tubular ER vs. sheets of ER is not supported in a convincing manner.

Previous work certainly has established that RTN3 functions in membrane tubulation and the role of lunapark in the organization of tubular membranes has also been made in other work. However, the confocal images in this manuscript do not clearly and independently justify the conclusions of tubular vs. sheet ER membrane localization. These claims should be justified by serial confocal or super resolution microscopy reconstruction of images to more clearly document the claim of tubular vs ER sheet localization.

*Reviewer #2 :*

Previous work from this group showed that some ER-phagy requires SEC24C and RTN3 (PMID: 31273116). The present study reveals these proteins are necessary for degradation of regions of the ER containing accumulations of Akita, a dominant interfering mutant of proinsulin. It also shows that the puncta are probably protein condensates rather than aggregates as had been thought. The condensates become enlarged when SEC24C and RTN3 are depleted, consistent with the idea that Akita condensates are degraded by ER-phagy. This part of the study is well done and convincing. The study goes on to argue that ER structure plays an important role in targeting Akita condensates for degradation by ER-phagy; decreasing the abundance of tubular ER reduces Akita condensate degradation, while restoring ER tubules promotes degradation. The authors suggest ER tubulation promotes Akita condensate degradation by facilitating their interaction with SEC24C and by physically limiting condensate growth, preventing them from becoming too large for degradation by ER-phagy. If true, this is an important conceptual advance in our understanding of how ER shape promotes protein segregation, affects condensate formation, and plays a role in quality control of ER proteins. However, additional work is necessary to make a stronger case for the model.

Overall, this is a well-done, largely convincing study. It demonstrates that ER-phagy requiring SEC24C and RTN3 degrades Akita condensates in the ER. However, the idea that the extent of ER tubulation is a major determinant of condensate size or targeting to sites of ER-phagy requires additional evidence.

1. The only way ER structure is altered is knockout of Lunapark, which could affect ER function in ways beyond altering structure. The study would be stronger if it were shown that increasing ER sheets by another method affected Akita degradation similarly to knockout of Lunapark. For example, ER sheets could be induced by overexpression of Climp63 [PMID: 21111237]. It is not necessary to repeat all the work with the Lunapark knockout, but it should be shown that increasing ER sheets by another mechanism causes an increase the percent of cells with large Akita puncta (as in Figure 4A) and reduces Akita degradation in LAMP1-positive structures (as in Figure 5—figure supplement 3A,B)

2. Deletion of Lunapark could affect Akita condensates in ways unrelated to changes in ER tubulation. Condensate formation is probably largely determined by the concentration of Akita in the ER, which could increase if the volume of the ER is reduced in cells lacking Lunapark or if Akita abundance increases in these cells. ER shape might also affect condensate mobility in the ER, which could alter the rate of coalescence of small condensates into larger ones. It also possible that the number or distribution of SEC24C puncta is altered in cells lacking Lunapark, which could reduce degradation of akita condensates. In short, there are a number of ways deletion of Lunapark could affect condensate size that are unrelated to the amount of tubulation of the ER. The study would be stronger if these other possibilities were ruled out but, at a minimum, they should be acknowledged and discussed.

3. The idea that ER tubules restrict the size of condensates seems implausible. What would stop condensates from expanding along the length of the tubule?

4. I do not understand the claim only a small fraction of Akita puncta are positive for RTN3 or Sec61 (Figure 6D-G; Figure 5 supplement 3,F). In the images provided all Akita puncta look positive for both proteins. Shouldn't Sec61-β, which is widely used as a general ER marker, be present in all ER domains together with Akita puncta?*Reviewer #3:*

The manuscript by Parashar et al., builds on previous work from the same group demonstrating the involvement of COPII coat proteins in selective routing of unfolded proteins in the ER for autophagic degradation by ER-phagy (Cui et al., 2019). While previous work has primarily been done in yeast, Parashar et al., now extends to mammalian cells (human) and focuses on mutated prohormones, especially Akita insulin, as cargo. Using human osteosarcoma U-2 OS cells for ectopic expression of an array of fluorescently tagged proteins, the authors demonstrate that manipulation of autophagy flux increases delivery of Akita insulin to LAMP1 lysosomes and this is dependent on cooperative interactions between SEC24C and the tubular ER-phagy receptor RTN3. Distinct pools of small and large Akita puncta are identified with the former associated with the tubular ER-network and the latter linked to sheet-like ER. Disruption of the tubular network in LNPK KO cells leads to larger puncta formation and this is partially rescued with RTN3 overexpression. A similar proposition is made for AVP and POMC prohormones carrying misfolding mutations.

The manuscript is well written and easy to follow, albeit could use improvement in reporting which expression system is used for each experiment (transient vs stable) and data quantification. Specific comments follow below.

– The use of fluorescently tagged proteins to avoid staining artifacts while easily allowing spatiotemporal tracking are seen as a strength of the study. This point would have been strengthened even further if authors provide a quantitation of the expressed proteins (SEC24C, RTN3, etc) relative to endogenous levels and avoid uncertainty due to exceedingly high ectopic expression. Using COL1A1 as an alternative cargo channeled through FAM134B helps increasing confidence about the selectivity of the SEC24C-RTN3 system.

– While it is acceptable to work out some of the mechanistic details in the chosen cellular system (U-2 OS), the physiological relevance of the study is clouded by not including cell lines that naturally process the studied cargos in their native state. The lack of a reference point (endogenous insulin) makes it very hard to interpret if some of the alterations in Akita puncta dynamics occur due to exceedingly high ectopic expression. In this sense, using murine MIN6 or human EndoC-βH1 to demonstrate SEC24C-RTN3-dependent Akita insulin ER-phagy degradation, while WT proinsulin is also present at comparable levels, would be a great addition to the study. This is also important since, as stated in the discussion (Page 14, lines 21-23), Akita forms oligomeric complexes with WT insulin. Along the same line, neuronal cell lines should be considered for POMC/AVP studies.

– Still on the physiological relevance, the Akita mouse is a model of progressive β-cell loss due to increased ER-stress causing overt diabetes (PMID: 11854325). Is the transcriptional ER-stress response further exacerbated by preventing ER-phagy degradation?

– Torin2 as a model of mTORC-regulated autophagy (Figure 1). In addition to inhibition of both mTORC1 and mTORC2, torin2 also exerts potent inhibitory effects against other kinases including ATR, ATM and DNA-PK (PMID: 2343680). While mTORC1/2 act as nutrient sensors, these other kinases are activated in response to DNA-damage and also regulate autophagy (PMID: 31983282; PMID: 31911943).

– Is there a context selectivity (e.g. nutrient starvation, DNA-damage) that governs SEC24C-linked ER-phagy?

– Can SEC24C-linked ER-phagy be demonstrated in more physiologic models (e.g. serum withdrawal)?

– Seems like Torin2-induced autophagy is only used to demonstrate SEC24C colocalization with RTN3 and LC3B (Figure1) and only blockade of autophagic flux with BafA and inhibition of ULK activity with MRT68921 is used onwards. Is there a specific reason for that? Does Akita trafficking to LAMP1 lysosomes also increase in Torin2-treated cells or some of the conditions suggested above?

– Only a fraction of Akita insulin seems to undergo lysosomal degradation (Figure 2A). What is the fractional Akita content that is transported to LAMP1 lysosomes? It is not clear what "relative pixel intensity" means (Figure 2B). It would also be important that authors demonstrate the distribution of Akita molecules across other cell compartments (e.g. ER, Golgi) using specific markers.

– Page 8, Lines 11-12. "Akita puncta are not aggregates as previously suggested, but instead behave as liquid condensates". This is a strong proposition that should be consolidated with additional experiments. Proteins that form phase separated condensates show dose-dependency and are sensitive to increases in NaCl concentration in vitro (PMID: 32895492). Can this be demonstrated for the Akita insulin? Is this modified by the presence of WT proinsulin? If not further developed, authors should reconsider the use of "condensates" term in the title of the article. In addition, it is unclear why the photobleaching experiments in Figure 3F are done only MRT68921-treated cells and untreated controls are not included.

– The experiments in Figure 6 are difficult to interpret since the overexpression levels achieved for RTN3 and RTN4 are not reported.

– Introduction. Page 3, line 12. Please define HSAN.

– Results. Page 6, line 22-23. "Furthermore, large Akita puncta accumulated in cells depleted of SEC24C". Was this due to enlargement of small Akita puncta or did the total number of Akita puncta also increased by SEC24C depletion?

– Results. Page 8, line 21. "Proinsulin localizes to the ER, ERS and Golgi". Not clear what criteria are used to reach this conclusion since figure only shows proinsulin and SEC24 colocalization.

– Results. Page 9, line 14. Quantitation of transfection controls (siCTL) is reported for experiments on Figure 4, but no representative images are provided in the main Figure or associated supplement.

– Results. Page 9, line 19-20. "…indicating that the association of RTN3 with LC3B does not depend on SEC24C". Could this be a compensatory effect of SEC23A? Is the Akita-LC3B colocalization preserved in double SEC24C/23A knockdown cells?

– Results. Page 10, line 18. "POMC-C28F is a mutant prohormone that causes early onset diabetes". In humans, the POMC-C28F variant has been linked to early onset obesity and not diabetes (PMID: 18697863).

[Editors’ note: further revisions were suggested prior to acceptance, as described below.]

Thank you for resubmitting your work entitled "Endoplasmic reticulum tubules limit the size of misfolded protein condensates" for further consideration by *eLife*. Your revised article has been evaluated by David Ron (Senior Editor) and a Reviewing Editor.

The manuscript has been improved but there are some remaining issues that need to be addressed, as outlined below:

You will see that reviewers #1 and 3 are fully supportive of this work but #2 has some remaining concerns which we feel may be addressed by additional measurements with data you should have in hand. Please address the concern in point 1 of reviewer #2. After further discussion, #2 offers the following comments on point 2:

The issue is that the images in 5D don't seem consistent with the quantification shown in 5G or I don't understand how the quantification was done. If they explain the quantification in more detail, it could be fine. Confirming the result by co-IP is not necessary if they quantification of the images is clear and the Dikic paper is cited.

Please address these concerns and we will make a final decision without the need to consult the reviewers again.

*Reviewer #1:*

I have reviewed the new version of the submission by Parashar et al. focusing on the comments I made in my review of the previous submission of this work. The authors have satisfied all my concerns. They have highlighted the results of the previously published work by Grumati et al. on the molecular association of RTN3 with SEC24C but not with the other paralogs of SEC24. Although they referenced this work in their previous submission, the inclusion of an explicit summary of those results helps to support the conclusion they wish to draw.

Two new results address my other concerns. They have now used reconstruction of serial Z-sections to document the association of large Akita puncta with sheets of ER in Lunapark KO cells and in cells over-expressing Climp63, but not when the over-expressed Climp63 is compensated by over-expression of RTN4. They have also demonstrated that the accumulation of COLA1 causes the accumulation of at least partially no-fluid domains of aggregate protein in contract the more fluid nature of Akita accumulated in ER tubules.

I see no other concerns and believe this interesting work merits publication in *eLife*.

*Reviewer #2:*

The revised version of this study is improved, but some issues remain. It now uses a second method to increase ER sheets (Climp63 overexpression) and finds this leads to the accumulation of large Akita puncta and decreases deliver of Akita to lysosomes. These changes are reversed by also overexpressing RTN4, which increases ER tubulation. Together, these results strengthen support for a correlation between the extent of ER tubulation and Akita condensate degradation rate. However, there are still concerns about how the microscopy results are quantified.

1. ER shape and condensate formation are quantified as "% cells with sheet-like ER" and "% cells with large Akita puncta" (Figure 5D, Figure 5 Sup1 CD). This makes it hard to see how well condensate abundance and size correlate with ER sheet volume. It would be better if the mean % of ER in sheets and the mean size of Akita condensates per cell were determined. This will help make it clearer that there is a good correlation between ER sheet abundance and condensate size and number.

2. The claim that Akita condensates are specifically enriched near RTN3 but not SEC61 is still not convincing. Rev 1 had related concerns (point 1). In response to my comments about how Akita colocalized with RTN3 is assessed, the rebuttal letter states "only RTN3 [and not SEC61] concentrated in puncta colocalize with the small Akita puncta in tubules" (Figure 6D), but Akita seems to be all over the ER, together with RTN3 and SEC61. This issue can be addressed in two ways. First, clearly define what counts as an Akita punctum (condensate) and what distinguishes them from the rest of the Akita signal. Second, use an objective measure of colocalization or association. Also, as Rev 1 suggests, it also would be good if condensate association with RTN3 were confirmed by a second method (e.g., by IPs as in ref 32).

*Reviewer #3:*

In their response letter and revised version of the manuscript "Endoplasmic reticulum tubules limit the size of misfolded protein condensates" by Parashar et al., the authors have responded to all points raised and have made modifications to improve data reporting and interpretation, especially regarding the expression levels of the cellular models used. More importantly, the authors now include new data on Akita puncta formation in insulin-secreting cell lines (INS1) and linked it to altered ER structure, which increases confidence about the physiological relevance of their findings making the manuscript suitable for publication.

---

## [Author Response]

[Editors’ note: the authors resubmitted a revised version of the paper for consideration. What follows is the authors’ response to the first round of review.]

Reviewer #1:Parashar et al. present interesting results on the organization of a specialized zone of the endoplasmic reticulum, ER tubules, that appear to organize the autophagic engulfment and diversion to the lysosome of a mutant, misfolded form of proinsulin, the Akita mutant protein. This specialized zone correlates to the localization of an ER membrane protein, RTN3, known to promote membrane tubulation and also to cytoplasmic subunits, SEC24C and SEC23A, of a coat protein complex that participates in sorting of membrane proteins destined to traffic the ER. In contrast to what had been assumed of aggregated forms of the Akita mutant proinsulin, the authors find that puncta of the accumulated protein remain in a fluid state, characteristic of protein condensates rather than of a more rigid protein aggregate.Parashar et al. have investigated the localization and fluid properties of mutant Akita that accumulates in the ER and is subject to a process of turnover in the lysosome termed ER-phagy. The work relies on live cell imaging of transfected forms of fluorescnety tagged forms of Akita, of wild type prolinsulin, collagen, RTN3, Sec24C and mutant forms of other prohormones. The broad conclusions are consistent with and extend previous observations on the role of RTN3 in formation of ER tubules in relation to the ER-phagy of other misfolded proteins subject to lysosomal degradation.The novel conclusions the authors make are that the Akita mutant protein is in a fluid protein condensate state rather than as an aggregate and that ER-phagy operates selectively on tubular membrane marked by their content of RTN3 in the membrane and SEC24C on the cytoplasmic face of the tubular membrane.I have several concerns that could be addressed by additional experimental work and a few small issues with the text.1. All of the conclusions of this work rely on visual inspection of fluorescently-tagged over-expressed proteins. At the very least, this should be expressed as a limitation of the analysis but it would help support the strength of the conclusions if at least some of the work was repeated by another approach. One such additional approach would be to use immunoprecipitation of RTN3 to detect a a possibly selective association with SEC24C under conditions that promote ER-phagy in non-transfected cells. If the endogenous level of these two proteins is too low, the authors might consider a proximity labeling approach with APEX-tagged Akita, RTN3 or SEC24C.

In support of our localization studies is an earlier study from Ivan Dikic’s lab that reported the interactome for RTN3L and RTN3S during starvation induced ER-phagy (Grumati et al. *eLife* 2017). The long form of RTN3 (RTN3L) is required for ER-phagy, while the short form (RTN3S) is not. RTN3L, but not RTN3S, specifically co-precipitated with the endogenous copy of SEC24C during starvation. Additionally, endogenous SEC24C, and not other SEC24 isoforms, co-precipitated with RTN3L. While the IP experiments with RTN3L and RTN3S were performed with stably-transfected cells, and the expression levels of ectopic RTN3L and RTN3S in these cell lines was not reported, the specificity of these IPs indicates that they are physiologically relevant. Additionally, all the co-precipitating proteins in the interactome were endogenously expressed proteins. These findings are now described in the text (see Results section entitled “SEC24C colocalizes with LC3B and RTN3 in Torin-treated cells”). Please note that our localization studies with RTN3 and SEC24C are totally consistent with what we observed in yeast when we analyzed the endogenous copy of these homologues (see Cui et al. Science 2019). This is also mentioned in the text. In addition, see our response to Reviewer #3 (first and second response) about the expression levels of the proteins we used in our studies, and the physiological relevance of our studies.

2. How is the Akita protein retained in proximity to RTN3 in the membrane? Is there a direct interaction or is the Akita condensate anchored by some other means to the putative tubular regions of the ER?

This is an interesting question. RTN3 does not have a luminal domain, therefore, it is unlikely that it binds directly to *Akita*. A possible scenario is that RTN3 interacts with *Akita* via a transmembrane chaperone that has a luminal domain. We plan to use a variety of approaches to address this point, but currently, a molecular answer to this question is beyond the scope of this manuscript.

3. The claim of tubular ER vs. sheets of ER is not supported in a convincing manner.Previous work certainly has established that RTN3 functions in membrane tubulation and the role of lunapark in the organization of tubular membranes has also been made in other work. However, the confocal images in this manuscript do not clearly and independently justify the conclusions of tubular vs. sheet ER membrane localization. These claims should be justified by serial confocal or super resolution microscopy reconstruction of images to more clearly document the claim of tubular vs ER sheet localization.

To show that the large *Akita* puncta accumulate in ER sheets, while the small puncta are present in tubules, we captured Z-slices and performed a 3D reconstruction of the *Akita* puncta from control and LNPK KO cells, as well as control cells transfected with Climp63 or Climp63 and RTN4 (see Figure 5B-C and movies S3 and S4; Figure 5—figure supplement 1 and movies S5 and S6). These data demonstrate that the large *Akita* puncta are present in the sheets, while the small puncta are in tubules. Representative images are shown in Figure 5B-C and Figure 5—figure supplement 1.

Reviewer #2:Previous work from this group showed that some ER-phagy requires SEC24C and RTN3 (PMID: 31273116). The present study reveals these proteins are necessary for degradation of regions of the ER containing accumulations of Akita, a dominant interfering mutant of proinsulin. It also shows that the puncta are probably protein condensates rather than aggregates as had been thought. The condensates become enlarged when SEC24C and RTN3 are depleted, consistent with the idea that Akita condensates are degraded by ER-phagy. This part of the study is well done and convincing. The study goes on to argue that ER structure plays an important role in targeting Akita condensates for degradation by ER-phagy; decreasing the abundance of tubular ER reduces Akita condensate degradation, while restoring ER tubules promotes degradation. The authors suggest ER tubulation promotes Akita condensate degradation by facilitating their interaction with SEC24C and by physically limiting condensate growth, preventing them from becoming too large for degradation by ER-phagy. If true, this is an important conceptual advance in our understanding of how ER shape promotes protein segregation, affects condensate formation, and plays a role in quality control of ER proteins. However, additional work is necessary to make a stronger case for the model.Overall, this is a well-done, largely convincing study. It demonstrates that ER-phagy requiring SEC24C and RTN3 degrades Akita condensates in the ER. However, the idea that the extent of ER tubulation is a major determinant of condensate size or targeting to sites of ER-phagy requires additional evidence.1. The only way ER structure is altered is knockout of Lunapark, which could affect ER function in ways beyond altering structure. The study would be stronger if it were shown that increasing ER sheets by another method affected Akita degradation similarly to knockout of Lunapark. For example, ER sheets could be induced by overexpression of Climp63 [PMID: 21111237]. It is not necessary to repeat all the work with the Lunapark knockout, but it should be shown that increasing ER sheets by another mechanism causes an increase the percent of cells with large Akita puncta (as in Figure 4A) and reduces Akita degradation in LAMP1-positive structures (as in Figure 5—figure supplement 3A,B)

We have now proliferated ER sheets by overexpressing Climp63 and shown that this leads to the accumulation of large *Akita* puncta in the ER (see Figure 5—figure supplement 1), and a decrease in the delivery of *Akita* to lysosomes (Figure 5—figure supplement 2A-B). Furthermore, we showed that the co-overexpression of Climp63 with RTN4, a protein that drives ER tubulation, decreased the number of large *Akita* puncta (Figure 5—figure supplement 1). Thus, by using two different methods to proliferate ER sheets, LNPK KO cells and Climp63 overexpression, we obtained the same outcome.

2. Deletion of Lunapark could affect Akita condensates in ways unrelated to changes in ER tubulation. Condensate formation is probably largely determined by the concentration of Akita in the ER, which could increase if the volume of the ER is reduced in cells lacking Lunapark or if Akita abundance increases in these cells. ER shape might also affect condensate mobility in the ER, which could alter the rate of coalescence of small condensates into larger ones. It also possible that the number or distribution of SEC24C puncta is altered in cells lacking Lunapark, which could reduce degradation of akita condensates. In short, there are a number of ways deletion of Lunapark could affect condensate size that are unrelated to the amount of tubulation of the ER. The study would be stronger if these other possibilities were ruled out but, at a minimum, they should be acknowledged and discussed.

While it is very difficult to address if the loss of LNPK reduces the volume of the ER, or changes the ER in ways that are unrelated to tubulation, we have performed several experiments to address the concerns stated above. First, we proliferated ER sheets by a second method, the overexpression of Climp63 (see new data added in Figure 5—figure supplement 1 to the Results section entitled “*Akita* puncta enlarge in sheet-like ER” and the discussion). Large *Akita* puncta accumulated in Climp63 overexpressing cells when ER sheets increased. The observation that two different methods, used to proliferate ER sheets, has the same outcome on *Akita* puncta size strengthens our proposal that *Akita* puncta enlarge in ER sheets. Second, we have shown that the number of SEC24C puncta is not reduced in LNPK KO cells (see Figure 5-supplement 6B and Results section entitled “The association of *Akita* with RTN3 is required for the formation of ERPHS”). Finally, we also measured the velocity of the *Akita* puncta in LNPK KO cells and found that the velocity of the small puncta was unaltered compared to control cells (Figure 5—figure supplement 7B and Results section “The association of *Akita* with RTN3 is required for the formation of ERPHS”). The velocity of larger *Akita* puncta is slightly increased in LNPK KO cells, but this was also observed for large puncta in cells when autophagosome formation was blocked with the ULK1/2 inhibitor, MRT68921 (Figure 3—figure supplement 1A). In total, our findings indicate that ER shape plays an important role in ER quality control.

3. The idea that ER tubules restrict the size of condensates seems implausible. What would stop condensates from expanding along the length of the tubule?

The rate of condensate expansion is proportional to the surface area that is accessible to the high MWT *Akita* oligomers in solution. In a sheet, a condensate would be accessible along its entire equatorial circumference. In a tubule, accessibility of the condensate would be limited to the ends facing the tubule lumen. A condensate in the tubule is inaccessible to the soluble *Akita* oligomers in the area adjacent to the wall. Thus, a condensate could expand along the length of a tubule, but its growth rate would be limited by its reduced accessibility. This is now explained in the discussion.

4. I do not understand the claim only a small fraction of Akita puncta are positive for RTN3 or Sec61 (Figure 6D-G; Figure 5 supplement 3,F). In the images provided all Akita puncta look positive for both proteins. Shouldn't Sec61-β, which is widely used as a general ER marker, be present in all ER domains together with Akita puncta?

The diffuse pool of *Akita* throughout the network colocalizes with RTN3 and SEC61 in the network, but only RTN3 concentrated in puncta colocalize with the small *Akita* puncta in tubules. SEC61 does not concentrate in puncta that colocalize with *Akita* puncta. This is now stated more clearly in the text.

Reviewer #3:The manuscript by Parashar et al., builds on previous work from the same group demonstrating the involvement of COPII coat proteins in selective routing of unfolded proteins in the ER for autophagic degradation by ER-phagy (Cui et al., 2019). While previous work has primarily been done in yeast, Parashar et al., now extends to mammalian cells (human) and focuses on mutated prohormones, especially Akita insulin, as cargo. Using human osteosarcoma U-2 OS cells for ectopic expression of an array of fluorescently tagged proteins, the authors demonstrate that manipulation of autophagy flux increases delivery of Akita insulin to LAMP1 lysosomes and this is dependent on cooperative interactions between SEC24C and the tubular ER-phagy receptor RTN3. Distinct pools of small and large Akita puncta are identified with the former associated with the tubular ER-network and the latter linked to sheet-like ER. Disruption of the tubular network in LNPK KO cells leads to larger puncta formation and this is partially rescued with RTN3 overexpression. A similar proposition is made for AVP and POMC prohormones carrying misfolding mutations.The manuscript is well written and easy to follow, albeit could use improvement in reporting which expression system is used for each experiment (transient vs stable) and data quantification. Specific comments follow below.

We now state in the Materials and methods (see section entitled “Image Analysis”) and/or in the legends when we used transient transfection or stable cell lines. Additional data quantification is also reported (see response below).

– The use of fluorescently tagged proteins to avoid staining artifacts while easily allowing spatiotemporal tracking are seen as a strength of the study. This point would have been strengthened even further if authors provide a quantitation of the expressed proteins (SEC24C, RTN3, etc) relative to endogenous levels and avoid uncertainty due to exceedingly high ectopic expression. Using COL1A1 as an alternative cargo channeled through FAM134B helps increasing confidence about the selectivity of the SEC24C-RTN3 system.

We agree that the use of fluorescently tagged proteins in live cells was the best method for the analysis of ERPHS in our studies. We have now quantitated expression levels for fluorescently tagged SEC24C, RTN3, LC3B and RTN4 in the Materials and methods (see section entitled “Image Analysis”). For markers that were introduced by transient transfection, there was no more than 2 fold average overexpression. For the mCherryRTN3 stable cell line there was approximately 4-5.5 fold overexpression. All localization experiments in the manuscript were performed by transient transfection, except for the experiments with RTN3, which were performed with stable cell lines as well as transient transfection. The results we obtained for RTN3 by both methods were the same, and except for Figure 1C-D, only the results with the stable cell line are shown in the manuscript (see legends for more info). Despite the ˜5 fold overexpression, the results we obtained with the stable RTN3 cell lines are physiologically relevant as we observed an increased colocalization of RTN3 with SEC24C in response to ER-phagy induction by Torin. This increased colocalization was specific for SEC24C, and was not seen with SEC24A, SEC13 and SEC16. Furthermore, it is of note that the RTN3-coat colocalization results that we report in this manuscript mimic what we observed in yeast with fluorescently tagged proteins expressed at the endogenous level (see Cui et al. Science 2019).

– While it is acceptable to work out some of the mechanistic details in the chosen cellular system (U-2 OS), the physiological relevance of the study is clouded by not including cell lines that naturally process the studied cargos in their native state. The lack of a reference point (endogenous insulin) makes it very hard to interpret if some of the alterations in Akita puncta dynamics occur due to exceedingly high ectopic expression. In this sense, using murine MIN6 or human EndoC-βH1 to demonstrate SEC24C-RTN3-dependent Akita insulin ER-phagy degradation, while WT proinsulin is also present at comparable levels, would be a great addition to the study. This is also important since, as stated in the discussion (Page 14, lines 21-23), Akita forms oligomeric complexes with WT insulin. Along the same line, neuronal cell lines should be considered for POMC/AVP studies.

We have performed several experiments to show that the cargos we analyzed in this study behave similarly to the way they would in insulin secreting cells or neuronal cell lines. First, we showed that when *Akita-sfGFP* and Proinsulin-FLAG are co-expressed in U2OS cells, they colocalize to the large *Akita* puncta (Figure 5—figure supplement 4). This observation is consistent with published studies showing that *Akita* (*Akita*-sfGFP) traps wild-type proinsulin in the ER in insulin secreting INS1 b cells (Haataja et al. J Biol Chem 2013 vol 288 p1896). Similar results were also obtained when G57S Pro-AVP-FLAG was co-expressed with Pro- Pro-AVP-HA, or C28F POMC-FLAG was co-expressed with POMCMyc (Figure 5—figure supplement 4).

We also performed experiments with insulin secreting cells. We decided not to use human EndoC-bH1 cells as it is difficult to obtain this cell line. Instead, we used INS1 cells which are regularly used by others to study *Akita*. While the compact nature of the INS1 cells made it difficult to resolve tubules vs sheets, we were able to test the most important conclusion in the manuscript (i. e. that *Akita* condensates enlarge in ER sheets). We proliferated sheets by overexpressing Climp63, in growth medium that stimulates insulin secretion, and showed this leads to a significant accumulation of large *Akita* puncta (Figure 5—figure supplement 5). Furthermore, 3D reconstructions of the ER suggested that the large puncta are in sheet-like ER. Please also see comment #1 of Reviewer #2 and my response.

– Still on the physiological relevance, the Akita mouse is a model of progressive β-cell loss due to increased ER-stress causing overt diabetes (PMID: 11854325). Is the transcriptional ER-stress response further exacerbated by preventing ER-phagy degradation?

An earlier study showed that *Akita*-myc expression, in HEK293T cells, increased XPB1 spicing, which was further increased by knocking down RTN3 (see Cunningham et al. Mol Cell vol 75, 2019). Given that *Akita* puncta enlarge in INS1 cells when ER sheets are proliferated by Climp63 overexpression, we anticipate that similar results would be obtained in an *Akita* mouse model.

– Torin2 as a model of mTORC-regulated autophagy (Figure 1). In addition to inhibition of both mTORC1 and mTORC2, torin2 also exerts potent inhibitory effects against other kinases including ATR, ATM and DNA-PK (PMID: 2343680). While mTORC1/2 act as nutrient sensors, these other kinases are activated in response to DNA-damage and also regulate autophagy (PMID: 31983282; PMID: 31911943).– Is there a context selectivity (e.g. nutrient starvation, DNA-damage) that governs SEC24C-linked ER-phagy?– Can SEC24C-linked ER-phagy be demonstrated in more physiologic models (e.g. serum withdrawal)?

We do not know if Torin is inhibiting other kinases, besides mTORC, but we used Torin in Figure 1 and Figure 1-supplement 1 (instead of starvation) for a reason. The goal was to compare the localization of SEC24C, during ER-phagy induction, to its yeast homologue Lst1. For the Lst1 localization studies (Cui et al. 2019 Science), ER-phagy was induced with a Tor inhibitor, and not starvation.

In this manuscript, we also induced SEC24C-mediated ER-phagy in ways that do not depend on Torin. Specifically, we used several misfolded proteins, *Akita*, G57S Pro-AVP and C28F POMC to induce ER-phagy. Additionally, it should be noted that RTN3mediated ER-phagy was previously shown to be induced by starvation (Grumati et al. *eLife* 2017).

– Seems like Torin2-induced autophagy is only used to demonstrate SEC24C colocalization with RTN3 and LC3B (Figure1) and only blockade of autophagic flux with BafA and inhibition of ULK activity with MRT68921 is used onwards. Is there a specific reason for that? Does Akita trafficking to LAMP1 lysosomes also increase in Torin2-treated cells or some of the conditions suggested above?

As mentioned above, we only used Torin in U2OS cells to compare our localization studies with yeast. The focus of this manuscript is not how SEC24C-mediated ER-phagy behaves in response to Torin or starvation. Rather, the focus is how RTN3-SEC24C mediated ERphagy and ER tubules contribute to ER quality control.

– Only a fraction of Akita insulin seems to undergo lysosomal degradation (Figure 2A). What is the fractional Akita content that is transported to LAMP1 lysosomes? It is not clear what "relative pixel intensity" means (Figure 2B). It would also be important that authors demonstrate the distribution of Akita molecules across other cell compartments (e.g. ER, Golgi) using specific markers.

Approximately 1-2% of the total *Akita* is delivered to the lysosome during ER-phagy. To estimate the fraction of *Akita* that is transported to lysosomes during ER-phagy, we divided the *Akita*-positive area that overlaps with LAMP1 by the total *Akita* area in the cell on a per-cell basis. The calculation for the total *Akita* in the cell included the *Akita* present in the network as well as the puncta. This information has now been added to the Materials and methods (see section entitled “Measurement of *Akita* delivery to lysosomes”).

As described in the Materials and methods (see section entitled “Measurement of *Akita* delivery to lysosomes”), and now the legend of Figure 2 (to make it more evident), the relative pixel intensity for each condition is the mean intensity of *Akita*-sfGFP in the pixels that overlap with LAMP1. The data was normalized to the mean intensity obtained in the DMSO control.

Peter Arvan’s lab has extensively studied the localization of Proinsulin and *Akita* in insulin secreting rat cells and other cell lines, including U2OS cells. He has shown that Proinsulin has a juxtanuclear localization that colocalizes with the early Golgi marker, p115, in U2OS cells (see Haataja et al. J Biol Chem 2013). We have also observed Proinsulin near the nucleus (see the first paragraph of the Results section entitled “*Akita* colocalizes with SEC24C-SEC23A and RTN3-LC3B”). In addition, previous studies have shown that *Akita* is retained in the ER in insulin and non-insulin secreting cells and is not in a juxtanuclear pool (Haataja et al. J Biol Chem 2013). We have observed a similar localization of *Akita* in U2OS cells and shown that the diffuse pool of *Akita* in the ER network colocalizes with ER markers (see Figure 5). We also report small *Akita* puncta in the ER tubules that colocalize with RTN3 puncta (see Figure 5D-E). The puncta that contain *Akita* and RTN3 are putative sites of ER-phagy. SEC61 does not concentrate in puncta that colocalize with *Akita* puncta.

– Page 8, Lines 11-12. "Akita puncta are not aggregates as previously suggested, but instead behave as liquid condensates". This is a strong proposition that should be consolidated with additional experiments. Proteins that form phase separated condensates show dose-dependency and are sensitive to increases in NaCl concentration in vitro (PMID: 32895492). Can this be demonstrated for the Akita insulin? Is this modified by the presence of WT proinsulin? If not further developed, authors should reconsider the use of "condensates" term in the title of the article. In addition, it is unclear why the photobleaching experiments in Figure 3F are done only MRT68921-treated cells and untreated controls are not included.

The criteria we have used to conclude *Akita* puncta are condensates are those established in Banani et al. 2017 *Nat Rev Mol Cell Biol* 18, 285-298. This is now explained more clearly in the text (see last paragraph of Results section entitled “SEC24C colocalizes with small, highly mobile *Akita* puncta”). We also consulted with experts in the condensate field (Michael Rosen, UT Southwestern; James Shorter, U of Penn) to ensure we were interpreting the literature correctly. For these reasons, we feel it is appropriate to use the term “condensates” in the title.

We performed photobleach experiments using a variety of conditions. For example, MRT68921-treated cells, LNPK KO cells and siRTN3 depleted cells (Figure 3-supplement 1, Figure 5-supplements 8 and 9) were used for the FRAP experiments. We used MRT68921-treated cells for the FRAP studies because it increased the number of large *Akita* puncta, which facilitated the FRAP analysis of these puncta. The large puncta that form in MRT68921-treated cells do not appear to behave differently from the large puncta in control cells (i. e. puncta movement was the same in the absence and presence of MRT68921-see Figure 3-supplement 1A). Furthermore, the FRAP data of the large puncta in LNPK KO cells and siRTN3 cells was the same as with MRT68921-treated cells.

– The experiments in Figure 6 are difficult to interpret since the overexpression levels achieved for RTN3 and RTN4 are not reported.

The level of RTN3 and RTN4 overexpression are now reported in the Materials and methods (see section entitled “Image Analysis”). But, please note that we are intentionally overexpressing these proteins to change ER shape. The data in Figure 6C confirms that overexpressing RTN3 and RTN4 in LNPK KO cells changed the shape of the ER. The overexpression of the tubule forming protein, RTN4, is commonly used to convert ER sheets to tubules. The consequence of RTN3 overexpression was also examined in our studies as a control. As RTN3 is an ER-phagy receptor, that is needed to clear *Akita* from the ER, the overexpression of RTN3 should decrease the number of large *Akita* puncta. Although RTN3 is not a tubule forming protein, RTN3 overexpression increases the density of three-way conjunctions in the ER (see Wu and Voeltz Dev Cell 2021 vol 56, 52-66). In LNPK KO cells, where fewer junctions are present, RTN3 overexpression also drives the conversion of sheets to tubules.

– Introduction. Page 3, line 12. Please define HSAN.

This has been defined.

– Results. Page 6, line 22-23. "Furthermore, large Akita puncta accumulated in cells depleted of SEC24C". Was this due to enlargement of small Akita puncta or did the total number of Akita puncta also increased by SEC24C depletion?

This was due to the enlargement of the *Akita* puncta in the siSEC24C cells as the %cells with puncta, of all sizes, did not change when puncta were compared in siSEC24C vs siCtrl cells. These numbers are now stated in the legend to Figure 2.

– Results. Page 8, line 21. "Proinsulin localizes to the ER, ERS and Golgi". Not clear what criteria are used to reach this conclusion since figure only shows proinsulin and SEC24 colocalization.

This sentence has been changed. Please see the first paragraph in the Results section entitled “*Akita* colocalizes with SEC24C-SEC23A and RTN3-LC3B” for more information.

– Results. Page 9, line 14. Quantitation of transfection controls (siCTL) is reported for experiments on Figure 4, but no representative images are provided in the main Figure or associated supplement.

We have now provided a representative image of the colocalization of *Akita* with SEC24C and SEC24A at 0h of treatment with MRT68921 (see Figure 4-supplement 1A). Images for the 0h time point were not provided in the previous submission. We also provided an image for the colocalization of *Akita* with SEC24C in siRTN3 cells that were treated with MRT68921 for 3.5 h (Figure 4-supplement 3A). Images for the siCtl at 0h and 3.5h were not provided as they are completely redundant with the images in Figure 4-supplement 1A (0h MRT68921) and Figure 4A (3.5h MRT68921). We also did not provide an image for the colocalization of *Akita* with SEC24C in siRTN3 cells at 0h MRT68921 as it is redundant with Figure 4-supplement 3A.

Please note that we added a Ctrl image for *Akita*-LC3B colocalization to Figure 5—figure supplement 6D.

– Results. Page 9, line 19-20. "…indicating that the association of RTN3 with LC3B does not depend on SEC24C". Could this be a compensatory effect of SEC23A? Is the Akita-LC3B colocalization preserved in double SEC24C/23A knockdown cells?

We do not think there are compensatory effects of SEC23A. All SEC24 isoforms are found in a complex with SEC23. A key function of SEC23 in the COPII coat is to recruit the COPII coat outer shell, SEC31-SEC13, which does not appear to be required for ER-phagy in mammalian cells or yeast (see Cui et al. Science 2019). This observation implies that SEC23 associates with ERPHS, due to its stoichiometric association with SEC24C, and does not directly contribute to ER-phagy.

– Results. Page 10, line 18. "POMC-C28F is a mutant prohormone that causes early onset diabetes". In humans, the POMC-C28F variant has been linked to early onset obesity and not diabetes (PMID: 18697863).

This has been corrected.

[Editors’ note: what follows is the authors’ response to the second round of review.]

Reviewer #2:The revised version of this study is improved, but some issues remain. It now uses a second method to increase ER sheets (Climp63 overexpression) and finds this leads to the accumulation of large Akita puncta and decreases deliver of Akita to lysosomes. These changes are reversed by also overexpressing RTN4, which increases ER tubulation. Together, these results strengthen support for a correlation between the extent of ER tubulation and Akita condensate degradation rate. However, there are still concerns about how the microscopy results are quantified.1. ER shape and condensate formation are quantified as "% cells with sheet-like ER" and "% cells with large Akita puncta" (Figure 5D, Figure 5 Sup1 CD). This makes it hard to see how well condensate abundance and size correlate with ER sheet volume. It would be better if the mean % of ER in sheets and the mean size of Akita condensates per cell were determined. This will help make it clearer that there is a good correlation between ER sheet abundance and condensate size and number.

Please see Figure 5—figure supplement 1E and F for the new presentation of the data as the mean size of *Akita* puncta per cell that this reviewer requested. The following samples were analyzed: Control, CLIMP63 and CLIMP63 +RTN4 overexpressing cells (Figure 5—figure supplement 1E); Control, LNPK KO and LNPK KO+RTN4 overexpressing cells (Figure 5—figure supplement 1F). The explanation for how this data was calculated can be found in the Materials and methods in the section entitled “Image Analysis”.

We do not feel we can objectively calculate the mean % of ER in sheets per cell using the current images, as suggested in the decision letter. An objective calculation would likely require serial thin section electron microscopy or tomography. In brief, since we could not find examples in the literature on how to calculate the numbers the reviewer requested, we tried different thresholding methods to capture tubules and sheets. We then deleted by hand individual tubular objects, leaving only what we considered sheets. This process involved making many arbitrary decisions for each cell we quantitated. In the end, we found this method to be far less objective than the method that is currently reported in the manuscript. To do the analysis that is reported in the manuscript (% cells with mostly sheet-like ER), we used visual pattern recognition to bin cells as either mostly sheet-like or mostly tubular. This methodology is standard in the field, and routinely used in the laboratories of Tom Rapoport and Gia Voeltz (for an example see Figure 8B and legend in Wang et al. 2016 *eLife* 5:e18605).

2. The claim that Akita condensates are specifically enriched near RTN3 but not SEC61 is still not convincing. Rev 1 had related concerns (point 1). In response to my comments about how Akita colocalized with RTN3 is assessed, the rebuttal letter states "only RTN3 [and not SEC61] concentrated in puncta colocalize with the small Akita puncta in tubules" (Figure 6D), but Akita seems to be all over the ER, together with RTN3 and SEC61. This issue can be addressed in two ways. First, clearly define what counts as an Akita punctum (condensate) and what distinguishes them from the rest of the Akita signal. Second, use an objective measure of colocalization or association. Also, as Rev 1 suggests, it also would be good if condensate association with RTN3 were confirmed by a second method (e.g., by IPs as in ref 32).

In a cell, *Akita* is uniformly distributed throughout the ER tubules and sheets, as well as concentrated in puncta. To specifically quantify the *Akita* puncta in the network, we applied a threshold algorithm (YEN, in ImageJ) that identifies punctate accentuations in the network. We then asked if the identified *Akita* puncta colocalized with RTN3 or SEC61 puncta. To identify RTN3 or SEC61 puncta in the network, we used the same thresholding algorithm. We then identified colocalized puncta (*Akita*-RTN3, *Akita*-SEC61) using the Boolean image calculator in ImageJ. The data in Figure 5G was then calculated as follows: *Akita* puncta colocalized with RTN3 puncta / total *Akita* puncta X 100% or *Akita* puncta colocalized with SEC61 puncta / total *Akita* puncta X 100%. This information is now provided in the legend to Figure 5. Also see our comment about the Dikic paper in point #1 to Reviewer #1 in the last rebuttal letter.

Please note, in addition to referencing the revised Figures in the text, we also made a few small changes. Some of these changes include 1) revising the abstract, 2) The y-axis in Figure 5—figure supplement 1C, Figure 6C, and Figure 6—figure supplement 1 was changed to % Cells with mostly sheet-like ER, 3) The legend to Figure 5-source data 1 was revised, 4) A typo was corrected in the Materials and methods in the section entitled “Small interfering RNA knockdowns”.